# Trajectory-Aware Certified Decentralized Unlearning via SGD Stability

**Hengliang Wu** [* 1]   **Jiale Yang** [* † 1 2]   **Shuzhen Chen** [3]   **Di Wang** [4]   **Dongxiao Yu** [1]   **Youming Tao** [5]

## Abstract

Decentralized Unlearning (DU) aims to remove the influence of specific clients from a collaboratively trained global model. However, existing methods suffer from strong reliance on static, problem-specific hyperparameters or restrictive convexity assumptions, limiting their general applicability. To overcome these limitations, we propose **TRA**jectory-aware **CE**rtified **D**ecentralized **U**nlearning (**TRACE-DU**), a generic unlearning framework for decentralized training. **TRACE-DU** introduces a fine-grained sensitivity analysis that leverages local SGD updates and decentralized training dynamics, thereby eliminating the need for convexity assumptions and reducing dependence on manually tuned parameters. By integrating strategic checkpoint selection with calibrated noise perturbation, the proposed framework enables efficient certified unlearning. Moreover, we exploit historical model trajectories to extend this framework, enabling it to naturally support sequential unlearning requests from an arbitrary number of clients. We provide theoretical guarantees for certified unlearning and derive sensitivity bounds under both convex and non-convex loss functions. Experimental results demonstrate that our framework outperforms state-of-the-art baselines across diverse metrics.

## 1. Introduction

Collaborative learning methods, including Federated Learning (FL) (McMahan et al., 2017) and Decentralized Learning (DL) (Lian et al., 2017), have advanced privacy-preserving machine learning. However, these methods face a significant challenge in complying with new privacy regulations (e.g., GDPR) (Voigt and Von dem Bussche, 2017), which guarantee clients the right to be forgotten (RTBF). Specifically, clients may request the removal of private data at any time during training. While a straightforward solution is to retrain the model on the retained data after a deletion request, this is often prohibitively expensive. Consequently, Machine Unlearning (MU) has emerged as a promising solution to avoid the cost of retraining while fulfilling privacy requirements (Cao and Yang, 2015). The formal privacy guarantee for MU requires that the unlearned model be statistically indistinguishable from the one retrained from scratch. Inspired by differential privacy, Guo et al. (2020) introduced the definition of $(\epsilon, \delta)$-unlearning to guarantee model indistinguishability (i.e., certified unlearning).

However, implementing MU in collaborative learning poses additional challenges. Existing work has shown that deleted data can be inferred by adversaries from the trained model (Tao et al., 2024; Chen et al., 2025). Consequently, effective unlearning in FL or DL requires removing not only the data but also its influence on the consensus model. In a system with multiple clients, this can be translated to removing the impact of specific clients who request complete deletion (i.e., client-wise unlearning) (Zhong et al., 2025). Numerous efforts have been made to address machine unlearning in collaborative learning, but most of them focus on centralized settings with a central server (Liu et al., 2024).

The lack of a central coordinator prevents the straightforward application of centralized unlearning methodologies to decentralized regimes. Currently, only a handful of studies have addressed unlearning within fully decentralized settings. Despite these advancements, existing approaches are plagued by significant limitations, including lack of theoretical guarantees (Ye et al., 2024), heavy reliance on problem-specific parameters and convexity assumptions (Qiao et al., 2025a; Wu et al., 2026), and restrictions to fixed topologies (Lamri and Maniatakos, 2025). To address these challenges, we propose several key insights to advance certified and

---

*Equal contribution . † Work done during an internship at Shandong University. [1]School of Computer Science and Technology, Shandong University, Qingdao, China [2]School of Mathematics and Statistics, Central South University, Changsha, China [3]College of Computer Science and Technology, Ocean University of China, Qingdao, China [4]Provable Responsible AI and Data Analytics (PRADA) Lab, King Abdullah University of Science and Technology, Jeddah City, Saudi Arabia [5]School of Electrical Engineering and Computer Science, TU Berlin, Berlin, Germany. Correspondence to: Shuzhen Chen <szchen@ouc.edu.cn>, Di Wang <di.wang@kaust.edu.sa>, Dongxiao Yu <dxyu@sdu.edu.cn>, Youming Tao <tao@ccs-labs.org>.

*Proceedings of the 43rd International Conference on Machine Learning*, Seoul, South Korea. PMLR 306, 2026. Copyright 2026 by the author(s).

efficient decentralized unlearning.

**(1). Tracing sensitivity via decentralized training trajectories.** Beyond establishing model indistinguishability, certified unlearning requires rigorously bounding the error distance between the unlearned model and the exact retrained one. This requirement is technically known as *model sensitivity* bound. Model sensitivity is critical to certified unlearning since it directly determines the noise magnitude and therefore governs the attainable utility. Most existing studies derive sensitivity bounds that primarily depend on static problem-specific parameters (Sekhari et al., 2021; Lamri and Maniatakos, 2025; Wu et al., 2026). However, in decentralized scenarios, these static bounds fail to capture the complex dynamics characterizing how local updates contribute to the consensus model throughout the collaborative learning process. Such client-specific contributions can be naturally quantified by the deviation between clients' local models and consensus models along the training trajectory. Motivated by this observation, we introduce a fine-grained sensitivity analysis derived from the decentralized training dynamics and trajectory, which facilitates superior unlearning performance.

**(2). Unlocking the power of local SGD updates.** In conjunction with evaluating client contributions from a global view, we further investigate the process of local updates, which encompass local SGD computation and mixing aggregation between neighbors. While existing studies largely prioritize analyzing the influence propagation in peer-to-peer communication and mixing updates (Qiao et al., 2025a; Wu et al., 2026), we emphasize the significant potential of the local SGD step itself. To be specific, we take insights from the well-established stability of SGD (Hardt et al., 2016) and first leverage this property to achieve certified decentralized unlearning.

**(3). Going beyond the convexity setting.** Current certified unlearning methods often leverage second-order information for model correction (Sekhari et al., 2021; Suriyakumar and Wilson, 2022), a straightforward and effective approach that has been extended to decentralized settings (Wu et al., 2026). However, the theoretical guarantees for these methods typically rely on the convexity assumption, which often does not hold in practical scenarios such as neural networks (He et al., 2016). While empirical studies have demonstrated their effectiveness, we argue that a theoretical breakthrough beyond convexity assumptions is essential for generalizing certified decentralized unlearning.

**Our Contributions.** We propose a generic framework called **TRA**jectory-aware **CE**rtified **D**ecentralized **U**nlearning (**TRACE-DU**) to tackle the associated challenges in a unified, efficient manner. We start by leveraging both properties of local SGD update rules and individual contributions to establish a traceable sensitivity quantifica-

tion, which is based on the model update trajectory. We then utilize a sensitivity threshold to select the checkpoint and perturb it with carefully calibrated noise. Finally, we perform subsequent training initialized from the perturbed model. To better align with practical scenarios, we further extend our framework to handle sequential unlearning requests.

In summary, our main contributions are threefold:

- By leveraging model updates from a dual local-global perspective, we propose a generic certified DU framework under dynamic topologies with a traceable sensitivity bound and checkpoint selection. Notably, the proposed sensitivity bound relaxes the strict dependency on problem-specific parameters. Furthermore, our framework can be integrated into standard decentralized SGD frameworks and applies to both convex and non-convex optimization regimes.
- We provide a rigorous theoretical analysis demonstrating that our framework achieves $(\epsilon, \delta)$-unlearning. Moreover, we establish an explicit high-probability sensitivity bound with decentralized training dynamics. All our theoretical results are free from convexity assumptions, bridging a notable gap.
- Extensive experiments further confirm that our method outperforms state-of-the-art baselines across diverse metrics: model utility, unlearning quality, privacy protection, and unlearning efficiency.

## 2. Related Work

**Machine Unlearning** Machine Unlearning (MU) aims to efficiently remove the influence of specific data points from trained models without fully retraining. The first formal definition of MU was introduced by Cao and Yang (2015), which requires the outputs of an unlearning algorithm to match those of a model obtained through retraining (i.e., exact unlearning). Since then, various studies have proposed alternative definitions ensuring statistical indistinguishability between unlearned and retrained models (i.e., approximate unlearning) (Guo et al., 2020). A certified approximate unlearning algorithm should satisfy formal certified removal guarantees such as $(\epsilon, \delta)$-unlearning (Sekhari et al., 2021). Prior approaches often utilized the second-order information to scrub the trained model (Sekhari et al., 2021; Suriyakumar and Wilson, 2022) or used gradient-based methods for perturbation (Neel et al., 2021). These techniques are effective but rely heavily on the convexity of loss functions. Given the widespread application of neural networks and their non-convex nature (He et al., 2016; Chu et al., 2025; Wang et al., 2025), relaxing the convexity requirement is essential, while the absence of a unique global minimum in non-convex functions makes this particularly challenging. Prior research on certified unlearning in non-convex settings

*Table 1.* Comparison of existing decentralized unlearning algorithms

| ALGORITHM | GUARANTEE | CONVEX | NON-CONVEX | DYN. TOPOLOGY | TRACEABLE SEN. |
|---|---|---|---|---|---|
| HDUS (YE ET AL., 2024) | × | × | × | √ | × |
| PDUDT (QIAO ET AL., 2025A) | √ | √ | √ | √ | × |
| RR-DU (LAMRI AND MANIATAKOS, 2025) | √ | √ | √ | × | × |
| CDU (WU ET AL., 2026) | √ | √ | × | √ | × |
| **TRACE-DU/SDU (OUR WORK)** | √ | √ | √ | √ | √ |

Note that *DYN. TOPOLOGY* denotes the decentralized network with dynamic topologies; *TRACEABLE SEN.* denotes the sensitivity bound derived from model update trajectories rather than static problem parameters.

has yielded several effective methods, primarily through noise perturbation (Zhang et al., 2024; Chien et al., 2024; Qiao et al., 2025b) or by selecting checkpoints for subsequent training (Wang et al., 2023; Fraboni et al., 2024; Mu and Klabjan, 2025). However, these methods focus on centralized or single-node architectures, posing fundamental challenges to their application in decentralized scenarios.

**Decentralized Unlearning** Decentralized Unlearning (DU) aims to eliminate the impact of a specific client from a fully decentralized system (Qiao et al., 2025a). To the best of our knowledge, there are currently only a handful of DU algorithms. We discuss the representative ones as follows. Ye et al. (2024) introduced a DU mechanism that leverages distilled seed models to achieve exact unlearning. The lack of unlearning guarantee makes their algorithm theoretically unconvincing. Additionally, the involvement of model distillation presents challenges for integration with existing decentralized SGD (DSGD) algorithms (Lian et al., 2017; Koloskova et al., 2020) and introduces additional computational overhead. Qiao et al. (2025a) proposed the first provably approximate DU algorithm with certified unlearning guarantees. While PDUDT incorporates noise perturbation based on local model parameters, it suffers from a critical limitation that the sensitivity bound involves an excessive number of problem parameters, making it hard to quantify and theoretically insufficient (see Theorem 4.9 in (Qiao et al., 2025a)). Another existing certified DU algorithm, RR-DU (Lamri and Maniatakos, 2025), operates via a random walk mechanism that performs projected noisy gradient ascent, but it only considers decentralized networks with fixed topologies instead of dynamic ones. Recently, Wu et al. (2026) constructed corrective model updates with Newton-style approximations and proposed an efficient certified DU algorithm. Although this method is highly effective, its theoretical guarantees rely on the assumption of convexity, which limits its applicability to a broader range of problems (Wu et al., 2026). These limitations motivate us to develop a generic DU framework that fulfills all the following requirements: certified unlearning guarantee, quantifiable sensitivity bounds, broad applicability and unlearning efficiency. Table 1 presents a comparison of the proposed approach with existing DU methods across multiple dimensions.

## 3. Preliminaries

**Decentralized Learning** We consider a fully decentralized learning (DL) system operating over a network with potentially time-varying topology. Let $\mathcal{I}$ denote the set of indices of $N$ participating clients. Each client-$i$ ($i \in \mathcal{I}$) possesses its private local dataset $S_i$, which is drawn from a distinct local distribution $\mathcal{D}_i$. Unlike traditional FL, this system eliminates the central server and clients communicate exclusively with their current neighbors to train models collaboratively. Consistent with prior work (Qiao et al., 2025a; Wu et al., 2026), we also consider decentralized networks with dynamic topologies instead of fixed ones (Lamri and Maniatakos, 2025). This flexibility does not compromise our theoretical analysis, as shown in Appendix A.1. Specifically, the communication links between clients at round $t$ are captured by a time-varying symmetric doubly stochastic matrix $\mathbf{Q}^t \in \mathbb{R}^{N \times N}$, which means $\mathbf{Q}_{ij}^t = \mathbf{Q}_{ji}^t$ for all $i, j$ and $\sum_i \mathbf{Q}_{ji}^t = 1$ for all $j$. If $i$ and $j$ are not connected, $\mathbf{Q}_{ij}^t = 0$ (Lian et al., 2017). Furthermore, we make the following standard assumption of the spectral gap.

**Assumption 3.1.** Given the symmetric doubly stochastic matrix $\mathbf{Q}^t$, and let $\rho_k(\mathbf{Q}^t)$ denote the $k$-th largest eigenvalue of $\mathbf{Q}^t$. Define $\rho := (\max\{|\rho_2(\mathbf{Q}^t)|, |\rho_N(\mathbf{Q}^t)|\})^2$. We assume $\rho < 1$.

We denote $F_i(x, \xi_{n_i})$ as the training loss assessed on the data sample $\xi_{n_i}$ and $f_i(x) = \frac{1}{|S_i|} \sum_{n_i=1}^{|S_i|} F_i(x, \xi_{n_i})$ as the local function. Then we formalize the learning objective as the following decentralized optimization problem:

$$\min_{x \in \mathbb{R}^d} f_{\mathcal{I}}(x) = \frac{1}{N} \sum_{i=1}^{N} f_i(x). \tag{1}$$

During the learning process, clients collaboratively train a consensus model $\bar{x}^t = \frac{1}{N} \sum_{i=1}^{N} x_i^t$, where $x_i^t$ is the local model of client-$i$ after performing $t$ communication iterations. In each communication round $t$, client-$i$ conducts a local update by $K$ steps of SGD. We assume that the consensus model $\bar{x}^T$ converges to a stationary solution of the optimization problem (1) after $T$ rounds.

This process is similar to DSGD in (Koloskova et al., 2020) and is built on synchronous decentralized SGD, consistent

with standard decentralized optimization settings (Lian et al., 2017; Koloskova et al., 2020; Giladi et al., 2023). It is worth noting that the synchronization is a modeling assumption of the underlying training protocol instead of an additional requirement introduced by unlearning. In particular, our unlearning framework does not require extra synchronization beyond that already used in decentralized training.

To facilitate our analysis, we also make the following standard assumption regarding the loss functions. Our analysis accommodates a broad range of objective landscapes, including non-convex, convex, and $\lambda$-strongly convex settings.

**Assumption 3.2.** Loss functions $F_i(\cdot, \xi_{n_i})$ are $L$-smooth. Specifically, for any data sample $\xi_{n_i}$ and any $x, y \in \mathbb{R}^d$, we have $\|\nabla F_i(x, \xi_{n_i}) - \nabla F_i(y, \xi_{n_i})\|_2 \leq L \|x - y\|_2$.

**Decentralized Unlearning** Clients reserve the right to exit the system at any round $t$ ($t = 0, 1, \ldots, T$) and request that the models trained using their data be removed. To accommodate this, a decentralized unlearning algorithm is executed to generate an unlearned model that satisfies unlearning guarantees.

To theoretically guarantee the effectiveness of unlearning, the unlearned model should be statistically indistinguishable from a model trained without using the deleted data. This requirement is in line with the principles of certified approximate unlearning. More specifically, certified decentralized unlearning should satisfy the notion of $(\epsilon, \delta)$-unlearning which is formalized as follows. We clarify that since our focus is on client-wise unlearning (Zhang et al., 2023; Qiao et al., 2025a), the following definitions are adapted from the original $(\epsilon, \delta)$-unlearning (Sekhari et al., 2021). We provide both the standard client-wise $(\epsilon, \delta)$-unlearning definition and its high-probability version as follows.

**Definition 3.3** (Client-wise $(\epsilon, \delta)$-unlearning). Let $\mathcal{W} \subseteq \mathbb{R}^d$ denote the model space and $\mathcal{I}_-$ denote the retained client set obtained by deleting several clients. A learning algorithm $\mathcal{A}$ and an unlearning algorithm $\mathcal{M}$ achieve client-wise $(\epsilon, \delta)$-unlearning if, for any subset $W \subseteq \mathcal{W}$, we have

$$\mathbb{P}(\mathcal{M}(\mathcal{X}) \in W) \leq e^\epsilon \mathbb{P}(\mathcal{A}(\mathcal{I}_-) \in W) + \delta,$$
$$\mathbb{P}(\mathcal{A}(\mathcal{I}_-) \in W) \leq e^\epsilon \mathbb{P}(\mathcal{M}(\mathcal{X}) \in W) + \delta,$$

where $\mathcal{X}$ denotes the data statistics available to $\mathcal{M}$.

**Definition 3.4** (High-probability version). A learning algorithm $\mathcal{A}$ and an unlearning algorithm $\mathcal{M}$ achieve high-probability client-wise $(\epsilon, \delta)$-unlearning if there exist $\delta_G, \beta \in (0, 1)$ satisfying $\delta_G + \beta \leq \delta$, such that for any retained client set $\mathcal{I}_-$ and any subset $W \subseteq \mathcal{W}$,

$$\mathbb{P}(\mathcal{M}(\mathcal{X}) \in W) \leq e^\epsilon \mathbb{P}(\mathcal{A}(\mathcal{I}_-) \in W) + \delta_G + \beta, \quad (2)$$
$$\mathbb{P}(\mathcal{A}(\mathcal{I}_-) \in W) \leq e^\epsilon \mathbb{P}(\mathcal{M}(\mathcal{X}) \in W) + \delta_G + \beta. \quad (3)$$

Here, $\beta$ denotes the failure probability of the high-probability sensitivity bound. In particular, when $\delta_G +$

*Table 2.* Three cases of the stability factor $G(f_i, \gamma)$.

| CONVEXITY | LEARNING RATE | $G(f_i, \gamma)$ |
|---|---|---|
| NON-CONVEX | – | $1 + \gamma L$ |
| CONVEX | $\gamma \leq 2/L$ | $1$ |
| $\lambda$-STRONGLY CONVEX | $\gamma \leq 2/(L+\lambda)$ | $1 - \dfrac{\gamma L \lambda}{L + \lambda}$ |

$\beta \leq \delta$, the above guarantee reduces to client-wise $(\epsilon, \delta)$-unlearning.

Additionally, for any given client-$c$ ($c \in \mathcal{I}$) who requests deletion, we define the retained client set as $\mathcal{I}_c := \mathcal{I} \setminus \{c\}$. Similar to prior studies (Sekhari et al., 2021; Fraboni et al., 2024), we introduce the model sensitivity with respect to client-$c$ after $t$ rounds as

$$\mathcal{S}(t, c) := \|\bar{\omega}^t - \bar{x}^t\|_2, \quad (4)$$

where $\bar{x}^t$ is the consensus model obtained by applying learning algorithm for $t$ iterations over the data of clients in $\mathcal{I}$ and $\bar{\omega}^t$ is the consensus unlearned model for set $\mathcal{I}_c$. The sensitivity must be strictly bounded to guarantee the utility of the unlearned model. Furthermore, this sensitivity bound is further employed to calibrate the Gaussian noise required for achieving $(\epsilon, \delta)$-unlearning (Dwork and Roth, 2014; Sekhari et al., 2021).

## 4. Our Method

### 4.1. Trajectory-Aware Decentralized Unlearning via SGD Stability

**Sensitivity Bound with Decentralized Mixing.** In the training phase, each client first executes local SGD updates and then performs a weighted aggregation of neighboring models according to the mixing matrix. This mechanism motivates us to investigate the underlying dynamics of these local optimization trajectories. We begin by leveraging the expansion properties of stochastic gradient descent to analyze the local updates of client-$i$ (Hardt et al., 2016). Let the one step SGD update operator be denoted by $\mathcal{G}(f_i, \gamma, x_i^{t,k}) = x_i^{t,k} - \gamma \nabla F_i(x_i^{t,k}, \xi_i^{t,k})$, we then obtain:

$$\|\omega_i^{t,k+1} - x_i^{t,k+1}\|_2 = \|\mathcal{G}(f_i, \gamma, \omega_i^{t,k}) - \mathcal{G}(f_i, \gamma, x_i^{t,k})\|_2$$
$$\leq G(f_i, \gamma) \cdot \|\omega_i^{t,k} - x_i^{t,k}\|_2. \quad (5)$$

Here $x_i^{t,k}$ corresponds to the local model of client-$i$ at local step $k$ of round $t$ before unlearning, $\omega_i^{t,k}$ denotes the updated model after the removal of client-$c$. $G(f_i, \gamma)$ is a constant related to smoothness and convexity that reflects the stability of SGD. We list the three cases of $G(f_i, \gamma)$ in Table 2, which align with the result of Lemma 3.7 in (Hardt et al., 2016). Full details are provided in Appendix A.1.2.

Specifically, for a given client, the distance between the unlearned model and the original model is prevented from excessive expansion. Moreover, it is non-expansive under the convex setting and contractive when losses are strongly-convex, which guarantees the desired stability. However, it is infeasible to directly utilize Eq. (5) to bound sensitivity as we cannot directly access the unlearned model $\omega_i$.

For the next step, we introduce a proxy of client-specific contribution based on tracing decentralized training dynamics, which is further used to deliver the sensitivity bound. To begin with, decentralized learning dynamics involve the interplay of local SGD updates, neighbor aggregation, and global model mixing. Building upon the analysis of local SGD, we proceed to examine neighbor aggregation and global mixing.

For the neighbor aggregation process, the properties of the mixing matrix ensure that the peer-to-peer influence weights asymptotically converge to a uniform distribution as iterations progress. To be more specific, we have the following mixing bound to quantify how far the peer-to-peer influence at round $t$ deviates from the ideal uniform distribution:

$$||\Phi(t,p) - \frac{1}{N}\mathbf{1}_N\mathbf{1}_N^\top||_2^2 \leq \rho^{t-p+1}, \qquad (6)$$

where $\Phi(t,p) := Q^t Q^{t-1} \ldots Q^p$, $(t \geq p)$, $\mathbf{1}_N \in \mathbb{R}^N$ denotes the all-ones vector. For finite iterations where full consensus has yet to be attained, the influence weights remain only approximately uniform. As iterations progress and sufficient consensus is attained, the peer-to-peer influence weights can be effectively characterized by a uniform weighting scheme. Thus, in the well-mixed regime, we can use $\frac{1}{N-1}$ on the retained $N-1$ clients to approximate the peer-to-peer influence.

The global model mixing process implements distributed averaging over the communication graph, enabling clients to reach a consensus model via mechanisms like all-reduce protocols without central coordination, which is widely adopted in decentralized learning. From a dual local-global perspective, the divergence between locally available model $x_i^t$ and consensus model $\bar{x}^t$ (i.e., $||x_i^t - \bar{x}^t||_2$) can serve as a natural metric for the client-specific contribution along the training process. Integrating the aforementioned analysis, we construct a traceable and weighted proxy along the model update trajectory, formulated as:

$$\Omega_i(\mathcal{I}, \bar{x}^t) := \frac{1}{|\mathcal{I} \setminus \{i\}|}||x_i^t - \bar{x}^t||_2 = \frac{1}{N-1}||x_i^t - \bar{x}^t||_2. \tag{7}$$

The factor $\frac{1}{N-1}$ arises from the fact that the consensus model after the removal of client-$c$ can be given by $\bar{\omega}^t = \frac{1}{N-1}\sum_{i\in\mathcal{I}_c}\omega_i$. Additionally, if the consensus is sufficiently developed, the factor $\frac{1}{N-1}$ can also be viewed as the peer-to-peer influence weights since Eq. (6) implies that such

weights converge fast to uniform distribution. It is worth noting that $\Omega_i(\mathcal{I}, \bar{x}^t)$ closely mirrors the behavior of the consensus error in decentralized optimization. This structural similarity allows us to derive a computable, high-probability bound for $\Omega_i(\mathcal{I}, \bar{x}^t)$ by drawing upon the consensus error estimation and the following assumptions (Armacki and Sayed, 2026).

**Assumption 4.1.** We assume that the stochastic gradient noise $z_i^{p,k} := \nabla F_i(x_i^{p,k}, \xi_i^{p,k}) - \nabla f_i(x_i^{p,k})$ is light-tailed, i.e., $\mathbb{E}\left[\exp\left(\frac{||z_i^{p,k}||_2^2}{\sigma_i^2}\right)\Big|\mathcal{F}_{p,k}\right] \leq \exp(1)$, $\sigma_* := \max_{i\in\mathcal{I}}\sigma_i$.

**Assumption 4.2.** We assume that the local objectives satisfy the bounded heterogeneity condition $\max_{i\in\mathcal{I}}||\nabla f_i(x)||_2^2 \leq A^2 + B^2||\nabla f_{\mathcal{I}}(x)||_2^2$, $\forall i \in \mathcal{I}$, $\forall x \in \mathbb{R}^d$.

**Lemma 4.3** (High-probability bound for $\Omega_i(\mathcal{I}, \bar{x}^t)$). *Suppose Assumptions 3.1, 3.2, 4.1, and 4.2 hold. Let $H := Kt$ and define $\mathcal{R}_H(\eta) := \mathcal{O}\left(\frac{\sigma_*\sqrt{L}\left(\Delta_f + \log(2d/\eta)\right)}{\sqrt{NH}} + \frac{\Delta_f + \log(1/\eta)}{C_\alpha H} + \frac{N\rho L(A^2+\sigma_*^2)}{\sigma_*^2(1-\sqrt{\rho})^2 H}\right)$, where $C_\alpha > 0$ is the step size cap and $\Delta_f := f_{\mathcal{I}}(\bar{x}^0) - f_{\mathcal{I}}^*$. Let $W_t := \sum_{p=0}^{t-1}\rho^{(t-p)/2}$, and define $\mathcal{B}_\Omega(t,\beta) := \frac{8\gamma^2 K^2 N\sqrt{\rho}}{1-\sqrt{\rho}}\left[A^2 W_t + B^2 t\mathcal{R}_H(\frac{\beta}{2}) + t\sigma_*^2\left(1+\log\frac{2NH}{\beta}\right)\right]$. If the step size satisfies $\gamma \leq \min\left\{C_\alpha, \frac{\sqrt{N}}{\sigma_*\sqrt{15LH}}\right\}$ and $C_\alpha$ is sufficiently small, then for any fixed $i \in \mathcal{I}$ and any fixed $\beta \in (0,1)$, with probability at least $1 - \beta$, $\Omega_i(\mathcal{I}, \bar{x}^t) \leq \frac{1}{N-1}\sqrt{\mathcal{B}_\Omega(t,\beta)}$.*

Finally, we can establish the upper bound for model sensitivity, tying it to the stability of local SGD updates and client-specific contribution.

**Theorem 4.4.** *For smooth clients' local loss functions, let $G(f_{\mathcal{I}}, \gamma)$ denote $G(f_i, \gamma)$ for every $i \in \mathcal{I}$ and $c$ denotes the index of unlearned client, we have*

$$\mathcal{S}(t,c) \leq \sum_{p=0}^{t-1} G(f_{\mathcal{I}}, \gamma)^{(t-p-1)K} \cdot \Omega_c(\mathcal{I}, \bar{x}^p), \tag{8}$$

*where $\gamma$ is the learning rate, $K$ is the number of local training steps, and $\Omega_c(\mathcal{I}, \bar{x}^p) := \frac{1}{N-1}||x_c^p - \bar{x}^p||_2$.*

Compared with the centralized setting, the decentralized sensitivity bound involves an additional topology-dependent term induced by peer-to-peer mixing. In centralized learning, the global model is formed by exact server-side averaging, so the sensitivity recursion is governed only by the local SGD stability factor and the client contribution proxy, without any extra mixing discrepancy. Since the server directly collects all client updates and performs exact weighted aggregation, the contribution of client-$c$ can be explicitly quantified (Fraboni et al., 2024).

In contrast, in the decentralized setting, the consensus model is obtained only asymptotically through repeated communication over the network, and the influence of a client is

propagated through repeated peer-to-peer mixing. This is why the proxy $\Omega_i(\mathcal{I}, \bar{x}^t)$ could be further bounded through the network disagreement term and the mixing rate $\rho$ in Lemma 4.3. That is to say, the decentralized sensitivity bound inherits a consensus error term that is absent in centralized settings. Notably, the topology-dependent parameter $\rho$ explicitly appears in the sensitivity bound, capturing the finite-round discrepancy between decentralized mixing and ideal uniform averaging. The mixing bound in Eq. (6) shows that the discrepancy between finite-time decentralized mixing and ideal uniform averaging decays at rate $\rho^{\frac{t-p+1}{2}}$. In particular, a smaller $\rho$ implies faster consensus and a quicker decay of the mixing residual; whereas a larger $\rho$ indicates slower consensus and larger finite-round disagreement.

We emphasize that Lemma 4.3 provides a principled high-probability alternative for bounding the client-specific contribution term when the consensus model is not explicitly available to each client. In standard synchronous decentralized SGD, the consensus model can be computed via distributed averaging protocols such as All-Reduce, which do not rely on a central server and are commonly adopted in decentralized optimization systems (Lian et al., 2017; Giladi et al., 2023). Thus, in the following analysis and algorithmic description, we assume that clients can access the consensus trajectory through such standard synchronization protocols, while our high-probability result in Lemma 4.3 covers the alternative case where the consensus model is not available.

**Checkpoint Selection.** The sensitivity bound is further used to calibrate Gaussian noise and certified unlearning can be achieved by perturbing the model with this noise. Upon receiving a deletion request from client-$c$ at round $t_u$, a naive strategy is to directly perturb the consensus model at $t_u$ round with required noise and then continue training. However, it fails to address a critical trade-off inherent to the unlearning process. As the noise magnitude grows as training progresses, adding perturbation at round $t_u$ would severely damage model performance. Consequently, the subsequent recovery phase becomes prohibitively long to regain accuracy, compromising the efficiency of unlearning.

This motivates us to select an earlier checkpoint at which the required perturbation is smaller, while reducing the subsequent training cost to recover utility. To address the critical issue, we further leverage the sensitivity bound to select the unlearning time index. Specifically, we impose a threshold on the model sensitivity bound. Building upon the Gaussian mechanism (Dwork and Roth, 2014), the threshold is defined as $[2\ln(1.25/\delta)]^{-\frac{1}{2}} \epsilon\sigma$. Then we identify the most recent model time index $U$ (i.e., unlearning time index) which satisfies:

$$U = \arg\max_t (\Upsilon(t,c) \leq [2\ln(1.25/\delta)]^{-\frac{1}{2}} \epsilon\sigma), \quad (9)$$

where $\Upsilon(t,c) = \sum_{p=0}^{t-1} G(f_{\mathcal{I}}, \gamma)^{(t-p-1)K} \cdot \Omega_c(\mathcal{I}, \bar{x}^p)$. It can be updated by recurrence with $O(t)$ computational cost over $t$ rounds. After selecting the index $U$, we perturb the consensus model at round $U$ with the calibrated noise as $\tilde{x} = \bar{x}^U + \mathcal{N}(0, \sigma(U,c)^2 \mathbb{I}_d)$, where $\sigma(U,c) = \frac{\Upsilon(U,c)}{\epsilon} \cdot \sqrt{2\ln(1.25/\delta)}$. Then all retained clients continue training for $T_r$ rounds initialized from $\tilde{x}$, which is consistent with prior work (Qiao et al., 2025a). Algorithm 1 provides a detailed implementation of our framework.

Overall, **TRACE-DU** first maintains a fine-grained sensitivity bound grounded in the stability properties of SGD and a traceable proxy derived from locally accessible training information. Benefiting from this, we relax the reliance of the sensitivity bound on problem-specific parameters. Then this bound is utilized to identify the optimal checkpoint for injecting calibrated Gaussian noise.

Finally, leveraging the Gaussian mechanism, we establish $(\epsilon, \delta)$ unlearning guarantees for our framework.

**Theorem 4.5.** *The learning algorithm $\mathcal{A}$ and the unlearning algorithm $\mathcal{M}$ (**TRACE-DU**) satisfy $(\epsilon, \delta)$-unlearning of client-$c$ with Gaussian noise $\mathcal{N}(0, \sigma(t,c)^2 \mathbb{I}_d)$, where $\sigma(t,c) = \frac{\Upsilon(t,c)}{\epsilon} \cdot \sqrt{2\ln(1.25/\delta)}$ and $\Upsilon(t,c) = \sum_{p=0}^{t-1} G(f_{\mathcal{I}}, \gamma)^{(t-p-1)K} \cdot \Omega_c(\mathcal{I}, \bar{x}^p)$.*

**Corollary 4.6.** *Suppose the conditions in Lemma 4.3 and Theorem 4.4 hold. For a given deletion request from client-$c$, let $U$ denote the checkpoint time index selected for perturbation. Let $\{\beta_p\}_{p=0}^{U-1}$ be confidence parameters satisfying $\sum_{p=0}^{U-1} \beta_p \leq \beta$. Define the high-probability sensitivity bound $\widehat{\Upsilon}(U, c; \beta) := \sum_{p=0}^{U-1} G(f_{\mathcal{I}}, \gamma)^{(U-p-1)K} \cdot \frac{1}{N-1}\sqrt{\mathcal{B}_\Omega(p, \beta_p)}$. If **TRACE-DU** perturbs the selected checkpoint by Gaussian noise $\mathcal{N}(0, \sigma(U, c; \beta)^2 \mathbb{I}_d)$, where $\sigma(U, c; \beta) = \frac{\widehat{\Upsilon}(U,c;\beta)}{\epsilon}\sqrt{2\ln\frac{1.25}{\delta_G}}$, then the learning algorithm $\mathcal{A}$ and the unlearning algorithm $\mathcal{M}$ satisfy $(\epsilon, \delta_G + \beta)$-unlearning for client-$c$. In particular, by setting $\delta_G = \delta/2$ and $\beta = \delta/2$, **TRACE-DU** satisfies $(\epsilon, \delta)$-unlearning with $\sigma(U, c; \delta/2) = \frac{\widehat{\Upsilon}(U,c;\delta/2)}{\epsilon}\sqrt{2\ln\frac{2.5}{\delta}}$.*

### 4.2. Extension to Sequential Unlearning Requests

In what follows, we extend **TRACE-DU** to handle sequential unlearning requests from different clients. We denote $\mathcal{V}_{\mathcal{U}_{total}} = \cup_{u=0}^{\mathcal{U}_{total}} \mathcal{Q}_u$ as a series of $\mathcal{U}_{total}$ unlearning requests, where $\mathcal{Q}_0 = \emptyset$. Each unlearning request can contain an arbitrary number of clients. Let $\mathcal{I}_u := \mathcal{I} \setminus \mathcal{V}_u$ ($\mathcal{I}_0 = \mathcal{I}$) denote retained client set. As discussed before, we need to identify the unlearning time index $U$. However, this becomes challenging under sequential unlearning setting due to the difficulty in checking the precise time index across the whole unlearning sequence.

To address this challenge, we introduce *historical model*

**Algorithm 1 TRACE-DU**

---

**Input:** unlearning client index $c$, parameters $\epsilon, \delta, \sigma$, subsequent training steps $T_r$, initial local model $x_i^0 = \bar{x}^0$.

**Learning process:**
**for** each client-$i \in \mathcal{I}$ **in parallel do**
  **for** communication round $t = 0$ **to** $t_u - 1$ **do**
    **for** local update step $k = 0$ **to** $K - 1$ **do**
      $x_i^{t,k+1} \leftarrow x_i^{t,k} - \gamma \nabla F_i(x_i^{t,k}, \xi_i^{t,k})$;
    **end for**
    Client-$i$ sends $x_i^{t,K}$ to neighbors;
    Client-$i$ aggregates $x_i^{t+1} \leftarrow \sum_j \mathbf{Q}_{ij}^t x_j^{t,K}$;
    Client-$i$ computes
      $\Upsilon(t,i) \leftarrow \sum_{p=0}^{t-1} G(f_{\mathcal{I}}, \gamma)^{(t-p-1)K} \cdot \Omega_i(\mathcal{I}, \bar{x}^p)$;
  **end for**
**end for**
**Unlearning process:**
Client-$c$ submits unlearning request at round $t_u$;
Client-$c$ identifies the latest checkpoint
  $U = \arg\max_t \left( \Upsilon(t,c) \leq [2\ln(1.25/\delta)]^{-1/2} \epsilon\sigma \right)$;
Perturb the checkpoint as $\tilde{x} = \bar{x}^U + \mathcal{N}(0, \sigma(U,c)^2 \mathbb{I}_d)$;
**for** each retained client **do**
  Continue training for $T_r$ rounds initialized at $\tilde{x}$;
**end for**

---

*trajectory* $H$ to track and update the model history for each client. Each client can utilize $H$ to update its bounded sensitivity $\Upsilon(t,c)$ for each unlearning request. The selection of the unlearning index $U$ for a request $u$ depends on the past history of unlearning requests. Similarly to **TRACE-DU**, we identify the first unlearning index $U_1$ for the first unlearning request. Then we perturb the consensus model $\bar{x}_0^{U_1}$ and obtain the new initial model $\bar{x}_1^0$. Finally, we perform subsequent training of $T_r^1$ rounds and update the model trajectory as $H_1 = (\bar{x}_0^0, \ldots, \bar{x}_0^{U_1}, \bar{x}_1^0, \ldots, \bar{x}_1^{T_r^1})$. To handle the second unlearning request, we use the updated model trajectory $H_1$ to identify the second unlearning index $U_2$ and repeat the previous steps.

To be general, we define $H_u$ as the model trajectory after $u$ unlearning requests. For a set of clients $\mathcal{Q}_{u+1}$ in the next unlearning request, we need to compute the maximum bounded sensitivity according to $H_u$:

$$\Upsilon_{u+1}(t,c) := \sum_{p=0}^{t-1} G(f_{\mathcal{I}}, \gamma)^{(t-p-1)K} \cdot \Omega_{c,p}(\mathcal{I}_s, \bar{x}_s^p), \quad (10)$$

$$\Upsilon_{u+1}(t, \mathcal{Q}_{u+1}) := \max_{c \in \mathcal{Q}_{u+1}} \Upsilon_{u+1}(t,c), \quad (11)$$

where $\Omega_{c,p}(\mathcal{I}_s, \bar{x}_s^p)$ is the $p$-th element of the sequence $Seq_c^u = (\Omega_c(\mathcal{I}_s, \bar{x}_s^p)$ for $\bar{x}_s^p \in H_u)$ and $s = 0, 1, \ldots, u$.

After evaluating $\Upsilon_{u+1}(t, \mathcal{Q}_{u+1})$ along the model trajectory,

we can identify the optimal time index for perturbing as

$$U_{u+1} = \arg\max_t (\Upsilon_{u+1}(t, \mathcal{Q}_{u+1}) \leq [2\ln(1.25/\delta)]^{-\frac{1}{2}} \epsilon\sigma). \quad (12)$$

We proceed by perturbing the $U_{u+1}$-th model $H_u(U_{u+1})$ in $H_u$ with Gaussian noise defined in Theorem 4.5 to obtain the new initial model $\bar{x}_{u+1}^0$, where $\sigma(U_{u+1}, \mathcal{Q}_{u+1}) = \frac{\Upsilon_{u+1}(U_{u+1}, \mathcal{Q}_{u+1})}{\varepsilon} \sqrt{2\ln(1.25/\delta)}$. Then a subsequent training process is operated on retained clients to get $\bar{x}_{u+1}^{T_r^{u+1}}$ and to update the historical model trajectory as

$$H_{u+1} = (\bar{x}_0^0, \ldots, H_u(U_{u+1}), \bar{x}_{u+1}^0, \ldots, \bar{x}_{u+1}^{T_r^{u+1}}). \quad (13)$$

Our method requires storing $O(|H_u|d)$ historical parameters, where $|H_u|$ is the number of retained checkpoints and $d$ is the model dimension. By contrast, PDUDT requires each client to store its own local models and the models received from neighbors across rounds before unlearning, leading to per-client overhead $O(t_u N_{max} d)$, where $N_{max}$ is the maximum number of neighbors (Qiao et al., 2025a). In practice, this overhead can be further reduced using checkpoint sparsification, low-rank compression, or sliding-window caching that keeps only recent consensus checkpoints. Since our method does not require storing every client's full local trajectory, the additional storage overhead is much smaller than that of PDUDT. To make this comparison more concrete, we also measured the extra storage overhead for historical parameters (size in bytes) in Appendix A.3.

Finally, we establish certified unlearning guarantees for the sequential extension **TRACE-SDU**.

**Theorem 4.7.** *The learning algorithm $\mathcal{A}$ and the unlearning algorithm $\mathcal{M}'$ (**TRACE-SDU**) satisfy $(\epsilon, \delta)$-unlearning for clients in sequential unlearning requests. When we utilize the high-probability sensitivity bound, we also have high-probability $(\epsilon, \delta)$-unlearning guarantee.*

Details of all our theoretical results are provided in Appendix A.1.

# 5. Experiments

## 5.1. Experimental Setup

**Datasets and Models.** We use standard benchmark datasets in our experiments, including MNIST (Lecun et al., 1998), Fashion-MNIST (Xiao et al., 2017) (denoted as F-MNIST), and CIFAR-10 (Krizhevsky, 2009). For the convex task, we employ Logistic Regression on the MNIST dataset. For non-convex tasks, we utilize the CNN model on the Fashion-MNIST dataset and the ResNet-18 model (He et al., 2016) on the CIFAR-10 dataset. To further test the scalability and versatility, we extend our evaluation to larger datasets and text tasks, specifically employing the SVHN (Netzer et al.,

**Algorithm 2 TRACE-SDU** (Sequential DU)

---

**Input:** unlearning client sets $\{\mathcal{Q}_u\}_{u=0}^{\mathcal{U}_{\text{total}}}$, parameters $\epsilon, \delta, \sigma$, subsequent training steps $\{T_r^u\}_{u=0}^{\mathcal{U}_{\text{total}}}$

**for** $u = 0$ **to** $\mathcal{U}_{total} - 1$ **do**

    Update $\mathcal{I}_{u+1} \leftarrow \mathcal{I}_u \setminus \mathcal{Q}_{u+1}$;

    Compute $\Upsilon_{u+1}(t, \mathcal{Q}_{u+1})$ with Eq. (11);

    Compute unlearning time index $U_{u+1}$ with Eq. (12);

    Perturb to get the new initial model: $\bar{x}_{u+1}^0 = H_u(U_{u+1}) + \mathcal{N}(0, \sigma(U_{u+1}, \mathcal{Q}_{u+1})^2 \mathbb{I}_d)$;

    Continue decentralized training for $T_r^{u+1}$ rounds starting from $\bar{x}_{u+1}^0$;

    Update historical model trajectory $H_{u+1}$ with Eq. (13)

**end for**

---

2011) and AG News benchmarks (Zhang et al., 2015). Due to space constraints, the additional results are provided in Appendix A.3.4.

**Baselines and Evaluation Metrics.** We compare our proposed framework with three existing baselines: *Retrain*, which means retraining from scratch with retained clients by D-PSGD (Lian et al., 2017), *PDUDT* (Qiao et al., 2025a) and *Certified Hessian-based DU Algorithm* (Wu et al., 2026). For brevity, we denote the last one as *CDU*. Following prior studies (Khalil et al., 2025; Qiao et al., 2025a; Wu et al., 2026), we employ the following metrics: *Retain Accuracy*, *Forget Accuracy*, measuring the model's performance on retained client set and unlearned client set, respectively. *Membership Inference Attack (MIA)* indicates the extent to which the data of unlearned clients remains recognizable in the model. *Unlearning Time* denotes the runtime required to perform the unlearning process.

**Implementation Details.** We consider a fully decentralized system with *50* clients and no central server under non-IID data distributions, where the connections between clients are randomly generated. For **TRACE-DU**, we focus on a single unlearning request issued by one randomly selected client. For **TRACE-SDU**, we assess performance under the sequential unlearning setting, where the experimental protocol includes 10 sequential requests with one client removed per request and 5 sequential requests with 3 clients removed per request. We refer to these two scenarios as *Single-DU* and *Seq-DU*, respectively. For *Single-DU*, the unlearning request is set to occur at round $t_u = 40$ and we also set the number of subsequent training rounds to be equal to $t_u$ (i.e., $T_r = 40$) for fairness. For *Seq-DU*, the first unlearning request is set to occur at round $t_1 = 40$. Following a consistent evaluation protocol, we allocate 40 rounds of subsequent training for every unlearning request, which implies that unlearning requests occur sequentially every 40 rounds. Under this setting, we do not assume that the training procedure has converged when an unlearning request arrives. This formulation captures a more realistic

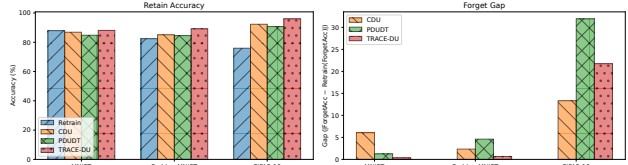

*Figure 1.* Comparison of retain accuracy and forget accuracy gap across different datasets under *Single-DU*.

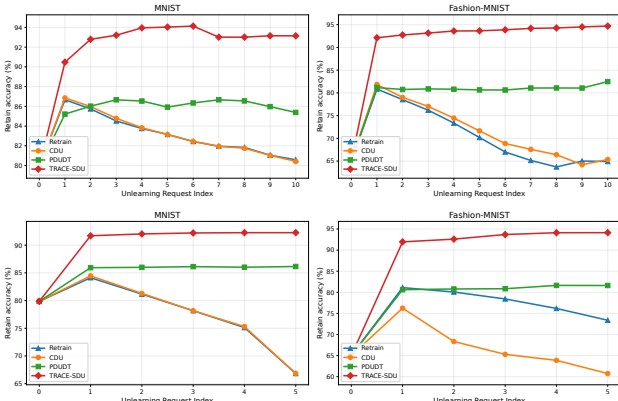

*Figure 2.* Comparison of the retain accuracy of different methods on different datasets under *Seq-DU*. *Top*: 10 sequential unlearning requests, removing one client per request. *Bottom*: 5 sequential unlearning requests, removing 3 clients per request.

scenario where clients may request deletion at any intermediate training stage, ensuring better compliance with privacy regulations.

Finally, to further substantiate the performance and robustness of our proposed algorithm, we have included comprehensive additional experiments. These evaluations encompass larger client scales, diverse network topologies, and sensitivity analyses regarding variations in $\sigma$, $\alpha$, $T_r$ and privacy budget. Specifically, to evaluate our framework under challenging conditions, we employ a different $\alpha$ and an expanded client base to evaluate robustness under different heterogeneity settings. Additionally, we simulate extreme high-frequency unlearning regimes by shortening $T_r$ and impose a tighter privacy budget through smaller values of $\epsilon$. We provide full details of our setup and additional experimental results in Appendix A.2 and A.3.

## 5.2. Unlearning Performance

**Model Utility** We measure the test accuracy on the retained clients to assess model utility (denoted as *Retain Accuracy*), where higher accuracy indicates better preservation of model performance. For *Single-DU* scenario, the results in Figure 1 demonstrate the improved performance of our proposed framework across different datasets. For *Seq-DU* scenario, we first ensure that all methods reach the same accuracy before the issuance of the first unlearning

Table 3. Comparison of MIA results on different datasets

| METHOD | MIA PRECISION (%) | | |
|---|---|---|---|
| | MNIST | F-MNIST | CIFAR-10 |
| RETRAIN | $56.31_{\pm 3.29}$ | $46.56_{\pm 3.00}$ | $51.63_{\pm 1.38}$ |
| PDUDT | $54.32_{\pm 1.61}$ | $49.60_{\pm 0.27}$ | $55.19_{\pm 1.44}$ |
| CDU | $54.37_{\pm 3.47}$ | $48.79_{\pm 1.09}$ | $50.12_{\pm 1.21}$ |
| **TRACE-DU** | $51.74_{\pm 0.82}$ | $49.22_{\pm 0.72}$ | $48.59_{\pm 2.30}$ |

Table 4. Unlearning time on different datasets

| METHOD | UNLEARNING TIME (S) | | |
|---|---|---|---|
| | MNIST | F-MNIST | CIFAR-10 |
| RETRAIN | 3136 | 2996 | 12290 |
| PDUDT | 6.23 | 11.23 | 18.02 |
| CDU | 4.91 | 8.92 | 20.84 |
| **TRACE-DU** | **0.02** | **0.11** | **2.21** |

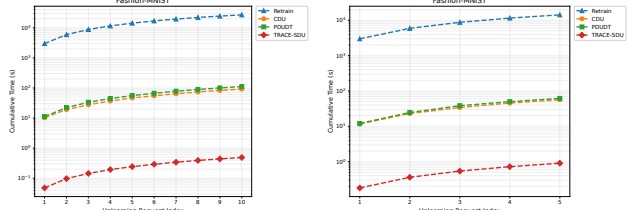

Figure 3. Comparison of cumulative unlearning time across different methods on the Fashion-MNIST dataset. *Left*: 10 sequential unlearning requests, removing one client per request. *Right*: 5 sequential unlearning requests, removing 3 clients per request.

request and denote the related unlearning request index as 0. We also reduce the interval between unlearning requests to a small number of rounds to characterize frequent sequential unlearning. Under this realistic setting, results in Figure 2 reveal that our framework maintains exceptional advantages over all baselines. Specifically, we record the retain accuracy after each method fully processes an unlearning request (i.e., the accuracy measured after the unlearning phase followed by subsequent training). As shown in Figure 2, all baselines suffer from performance degradation and fail to restore model utility during frequent sequential unlearning. In contrast, benefiting from our strategic checkpoint mechanism, **TRACE-SDU** consistently achieves superior accuracy within the same recovery budget. We provide more details of Figure 2 in Appendix A.2.4, including a justification for our method's performance advantage over *Retrain* baseline.

**Unlearning Quality**  We adopt the accuracy on the unlearned clients (forget set) as a primary metric for assessing unlearning quality (denoted as *Forget Accuracy*). Specifically, aligning with prior research (Khalil et al., 2025), we utilize the forget accuracy of the model retrained from scratch as the gold standard. Then we benchmark various methods by calculating the gap between their forget accuracy and this standard, where a minimal difference denotes superior unlearning quality. The results in Figure 1 demonstrate the effectiveness of our unlearning framework. Finally, we argue that forget accuracy alone is insufficient to fully characterize unlearning quality. Membership Inference Attack (MIA) metrics are also commonly employed for this evaluation, which we will discuss in the following part.

**Privacy Protection**  Consistent with prevailing research standards (Tao et al., 2024; Hong et al., 2024; Zhong et al., 2025; Qiao et al., 2025a; Li et al., 2025; Wu et al., 2026), we utilize MIA to assess the privacy protection performance of our framework. MIA aims to infer whether a specific data sample was part of the model's training set. In the context of unlearning, this entails assessing whether the target data can still be distinguished as a training sample by the unlearned model. Higher MIA precision implies severe privacy leakage, whereas the precision approaching 50% (equivalent to random guessing) indicates effective and secure unlearning. In our experiment, we conduct membership inference at-

tacks across different datasets under *Single-DU* scenario, reporting the average precision and standard deviation. As shown in Table 3, the MIA precision results of our method consistently approximate the random guessing. This implies that attackers cannot reliably distinguish training membership, thereby validating the privacy protection performance and unlearning quality of our unlearning framework.

**Unlearning Efficiency**  To evaluate the unlearning efficiency, we measure the runtime required for the unlearning process. Consistent with prior work (Qiao et al., 2025a), the reported runtime excludes training sessions before and after the unlearning request. Note that for *Retrain* method, the reported time reflects the full time of training from scratch, as it does not decouple unlearning from training. Table 4 reports the unlearning time for *Single-DU* scenario, while Figure 3 illustrates the cumulative unlearning time for *Seq-DU* scenario on F-MNIST datasets. Experimental results demonstrate that our algorithm not only maintains great unlearning performance but also exhibits superior efficiency.

## 6. Conclusion

In this paper, we propose a generic certified DU framework that exploits local SGD stability and a trajectory-based contribution proxy to derive a traceable sensitivity bound. This bound guides principled checkpoint selection and calibrated noise injection for efficient unlearning. We further extend it to handle sequential requests from arbitrary clients. We theoretically establish $(\epsilon, \delta)$-unlearning guarantees across diverse settings, with extensive experiments demonstrating superiority over state-of-the-art baselines.

## Impact Statement

This paper presents work whose goal is to advance the field of machine learning. There are many potential societal consequences of our work, none of which we feel must be specifically highlighted here.

## Acknowledgements

Shuzhen Chen was supported in part by the Postdoctoral Fellowship Program of the China Postdoctoral Science Foundation (CPSF) under Grant No. GZC20251040, in part by the Natural Science Foundation of Shandong Province under Grant No. ZR2025QC1537, in part by the Fundamental Research Funds for the Central Universities under Grant No. 202513024, and in part by the Postdoctoral Program of Qingdao under Grant No. QDBSH20250102012. Dongxiao Yu was supported in part by the Major Basic Research Program of the Shandong Provincial Natural Science Foundation under Grant No. ZR2025ZD18, and in part by the Joint Key Funds of the National Natural Science Foundation of China under Grant No. U23A20302. Di Wang is supported in part by the funding BAS/1/1689-01-01,RGC/3/7125-01-01, FCC/1/5940-20-05, FCC/1/5940-06-02, and King Abdullah University of Science and Technology (KAUST) – Center of Excellence for Generative AI, under award number 5940 and a gift from Google.

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

*Table 5.* Important notions used in our paper.

| Notations | Descriptions |
|---|---|
| $N$ | Total number of clients (nodes) in the decentralized network. |
| $i, j, c$ | Client indices; $c$ is the target client to unlearn. |
| $\mathbf{Q}_{ij}$ | Mixing weight between client-$j$ and $i$. |
| $\mathcal{I}$ | Total client set participating in training. |
| $\mathcal{I}_c$ | Retained client set after removing client $c$. |
| $t$ | Communication round index. |
| $t_u$ | Round when an unlearning request is issued. |
| $K$ | Number of local SGD steps per round. |
| $\gamma$ | Learning rate. |
| $f_i(\cdot)$ | Local function at client-$i$. |
| $\xi_{n_i}, \xi_i^{p,k}$ | Data sample at client-$i$. |
| $x_i^{t,k}$ | Local model of client-$i$ at round $t$ after $k$ local steps. |
| $x_i^t$ | Local aggregated model of client-$i$ at round $t$. |
| $\bar{x}^t$ | Consensus (average) model at round $t$. |
| $\tilde{x}$ | Perturbed checkpoint model (i.e., initial model for subsequent training). |
| $\Omega_c(\mathcal{I}, \bar{x}^t)$ | A proxy for the contribution of client-$c$ at round $t$. |
| $\Upsilon(t, c)$ | Sensitivity upper bound w.r.t. client-$c$ at round $t$. |
| $U$ | Selected checkpoint round in **TRACE-DU**. |
| $T_r$ | Number of subsequent training rounds after perturbation. |
| $\mathcal{Q}_u$ | Client subset removed in the $u$-th request. |
| $\mathcal{U}_{total}$ | Total number of sequential unlearning requests. |
| $\mathcal{I}_u$ | Retained client set after $u$ unlearning requests. |
| $U_u$ | Selected checkpoint for the $u$-th unlearning request. |
| $H_u$ | Historical model trajectory after $u$ unlearning requests. |
| $T_r^u$ | Number of subsequent training rounds for the $u$-th unlearning request. |

# A. Appendix

## A.1. Details of Theoretical Results

Here we deliver the notation table and full details of our theoretical results.

### A.1.1. PROOF OF LEMMA 4.3

*Proof.* We define $\Psi_i^t = \gamma \sum_{k=0}^{K-1} \nabla F_i(x_i^{t,k}, \xi_i^{t,k})$, $\mathbf{\Psi}^t = \mathrm{col}(\Psi_1^t, \ldots, \Psi_N^t)$, $\mathbf{x}^t = \mathrm{col}(x_1^t, \ldots, x_N^t)$ and $\bar{\mathbf{x}}^t$. Thus we have $x_i^K = x_i^t - \Psi_i^t$ and $\sum_{i \in \mathcal{I}} ||x_i^t - \bar{x}^t||^2 = ||\mathbf{x}^t - \bar{\mathbf{x}}^t||^2$. Let $J := \frac{1}{N} \mathbf{1}_N \mathbf{1}_N^\top$.

Next we bound $||\mathbf{x}^t - \bar{\mathbf{x}}^t||^2$ as follows. We first have

$$\mathbf{x}^t - \bar{\mathbf{x}}^t = ((\Phi(t-1, 0) - J) \otimes I_d)(\mathbf{x}^0 - \bar{\mathbf{x}}^0) + \sum_{p=0}^{t-1} ((\Phi(t-1, p) - J) \otimes I_d) \mathbf{\Psi}^p \tag{14}$$

and

$$||\mathbf{x}^t - \bar{\mathbf{x}}^t|| \le \rho^{\frac{t}{2}} ||\mathbf{x}^0 - \bar{\mathbf{x}}^0|| + \sum_{p=0}^{t-1} \rho^{\frac{t-p}{2}} ||\mathbf{\Psi}^p||. \tag{15}$$

From the geometric series estimation, we can obtain $\sum_{p=0}^{t-1} \rho^{\frac{t-p}{2}} \le \frac{\rho^{\frac{1}{2}}}{1-\rho^{\frac{1}{2}}}$. Then, based on the Cauchy-Schwarz Inequality $\left( \sum_p a_p b_p \right)^2 \le \left( \sum_p a_p \right) \left( \sum_p a_p b_p^2 \right)$ and $(a+b)^2 \le 2a^2 + 2b^2$, we have

$$||\mathbf{x}^t - \bar{\mathbf{x}}^t||^2 \le 2\rho^t ||\mathbf{x}^0 - \bar{\mathbf{x}}^0||^2 + \frac{2\rho^{\frac{1}{2}}}{1-\rho^{\frac{1}{2}}} \sum_{p=0}^{t-1} \rho^{\frac{t-p}{2}} ||\mathbf{\Psi}^p||^2. \tag{16}$$

Recall $\Psi_i^t = \gamma \sum_{k=0}^{K-1} \nabla F_i(x_i^{t,k}, \xi_i^{t,k})$ and $\mathbf{\Psi}^t = \text{col}(\Psi_1^t, \ldots, \Psi_N^t)$, we further have

$$||\mathbf{x}^t - \bar{\mathbf{x}}^t||^2 \leq 2\rho^t ||\mathbf{x}^0 - \bar{\mathbf{x}}^0||^2 + \frac{2\rho^{\frac{1}{2}}}{1 - \rho^{\frac{1}{2}}} \sum_{p=0}^{t-1} \rho^{\frac{t-p}{2}} \cdot 2\gamma^2 K \sum_{j \in \mathcal{I}} \sum_{k=0}^{K-1} ||\nabla F_j(x_j^{p,k}, \xi_j^{p,k})||^2. \tag{17}$$

Thus we get that

$$\frac{1}{N} \sum_{i \in \mathcal{I}} ||x_i^t - \bar{x}^t||^2 \leq \frac{1}{N} \left[ 2\rho^t ||\mathbf{x}^0 - \bar{\mathbf{x}}^0||^2 + \frac{4\gamma^2 K \rho^{\frac{1}{2}}}{1 - \rho^{\frac{1}{2}}} \sum_{p=0}^{t-1} \rho^{\frac{t-p}{2}} \sum_{j \in \mathcal{I}} \sum_{k=0}^{K-1} ||\nabla F_j(x_j^{p,k}, \xi_j^{p,k})||^2 \right]. \tag{18}$$

Let $D_t := \frac{1}{N} \sum_{i \in \mathcal{I}} ||x_i^t - \bar{x}^t||^2$. We also have $\Omega_i(\mathcal{I}, \bar{x}^t) = \frac{1}{N-1} ||x_i^t - \bar{x}^t|| \leq \frac{\sqrt{N}}{N-1} \sqrt{D_t}$. Finally, from Eq. (18), we get the desired bound of $\Omega_i(\mathcal{I}, \bar{x}^t)$, which is

$$\Omega_i(\mathcal{I}, \bar{x}^t) \leq \frac{1}{N-1} \left[ 2\rho^t ||\mathbf{x}^0 - \bar{\mathbf{x}}^0||^2 + \frac{4\gamma^2 K \rho^{\frac{1}{2}}}{1 - \rho^{\frac{1}{2}}} \sum_{p=0}^{t-1} \rho^{\frac{t-p}{2}} \sum_{j \in \mathcal{I}} \sum_{k=0}^{K-1} ||\nabla F_j(x_j^{p,k}, \xi_j^{p,k})||^2 \right]^{\frac{1}{2}}. \tag{19}$$

Assume $x_i^0 = \bar{x}^0$, we can simplify the bound as

$$\Omega_i(\mathcal{I}, \bar{x}^t) \leq \frac{1}{N-1} \left[ \frac{4\gamma^2 K \rho^{\frac{1}{2}}}{1 - \rho^{\frac{1}{2}}} \sum_{p=0}^{t-1} \rho^{\frac{t-p}{2}} \sum_{j \in \mathcal{I}} \sum_{k=0}^{K-1} ||\nabla F_j(x_j^{p,k}, \xi_j^{p,k})||^2 \right]^{\frac{1}{2}}. \tag{20}$$

We next further upper bound the gradient term in Eq. (20). For each local step $(p, k)$, define the stochastic-gradient noise as

$$z_i^{p,k} := \nabla F_i(x_i^{p,k}, \xi_i^{p,k}) - \nabla f_i(x_i^{p,k}). \tag{21}$$

We additionally use the following standard light-tailed noise and heterogeneity conditions: for all $i \in \mathcal{I}$, $p \geq 0$, $k \in \{0, \ldots, K-1\}$,

$$\mathbb{E}\left[ \exp\left( \frac{||z_i^{p,k}||_2^2}{\sigma_i^2} \right) \bigg| \mathcal{F}_{p,k} \right] \leq \exp(1), \qquad \sigma_* := \max_{i \in \mathcal{I}} \sigma_i, \tag{22}$$

and there exist constants $A, B \geq 0$ such that for all $x \in \mathbb{R}^d$ and all $i \in \mathcal{I}$,

$$||\nabla f_i(x)||_2^2 \leq A^2 + B^2 ||\nabla f_{\mathcal{I}}(x)||_2^2. \tag{23}$$

Here $f_{\mathcal{I}}(x) = \frac{1}{N} \sum_{i=1}^N f_i(x)$ is the global objective. The second condition is only used to convert the local-gradient energy into a problem-level high-probability envelope.

Let $H := Kt$ and confidence level $\eta \in (0, 1)$, with probability at least $1 - \eta$, we define

$$\mathcal{R}_H(\eta) := \mathcal{O}\left( \frac{\sigma_* \sqrt{L}(\Delta_f + \log(2d/\eta))}{\sqrt{NH}} + \frac{\Delta_f + \log(1/\eta)}{C_\alpha H} + \frac{N\rho L(A^2 + \sigma_*^2)}{\sigma_*^2 (1 - \sqrt{\rho})^2 H} \right), \tag{24}$$

where $C_\alpha > 0$ is the fixed stepsize cap, $\Delta_f := f_{\mathcal{I}}(\bar{x}^0) - f_{\mathcal{I}}^*$. The quantity $\mathcal{R}_H(\eta)$ is the non-convex high-probability gradient energy envelope obtained from Theorem 1 in (Armacki and Sayed, 2026) with the static spectral factor replaced by $\sqrt{\rho}$.

Specifically, under the corresponding stepsize condition

$$\gamma \leq \min\left\{ C_\alpha, \frac{\sqrt{N}}{\sigma_* \sqrt{15LH}} \right\}, \tag{25}$$

where

$$C_\alpha \le \min\left\{ \frac{1}{2L}, \frac{N}{9\sigma_*^2}, \frac{1-\sqrt{\rho}}{\sqrt{\rho}LB\sqrt{48}}, \frac{\sqrt{N}}{3\sigma_*\sqrt{5L}}, \frac{\sqrt[3]{N}(1-\sqrt{\rho})^{2/3}}{\sigma_*^{2/3}\rho L^{2/3}\sqrt[3]{9}} \right\}. \tag{26}$$

We have, with probability at least $1-\beta$,

$$\frac{1}{NH}\sum_{p=0}^{t-1}\sum_{k=0}^{K-1}\sum_{i=1}^{N}\|\nabla f_\mathcal{I}(x_i^{p,k})\|_2^2 \le \mathcal{R}_H(\beta). \tag{27}$$

Moreover, we already have

$$\|x_i^t - \bar{x}^t\|_2^2 \le \frac{4\gamma^2 K\sqrt{\rho}}{1-\sqrt{\rho}}\sum_{p=0}^{t-1}\rho^{\frac{t-p}{2}}\sum_{i=1}^{N}\sum_{k=0}^{K-1}\|\nabla F_i(x_i^{p,k},\xi_i^{p,k})\|_2^2. \tag{28}$$

For every $i,p,k$, we decompose $\nabla F_i(x_i^{p,k},\xi_i^{p,k}) = \nabla f_i(x_i^{p,k}) + z_i^{p,k}$. By $\|a+b\|_2^2 \le 2\|a\|_2^2 + 2\|b\|_2^2$ and the bounded heterogeneity condition,

$$\|\nabla F_i(x_i^{p,k},\xi_i^{p,k})\|_2^2 \le 2A^2 + 2B^2\|\nabla f_\mathcal{I}(x_i^{p,k})\|_2^2 + 2\|z_i^{p,k}\|_2^2. \tag{29}$$

Let

$$W_t := \sum_{p=0}^{t-1}\rho^{(t-p)/2} = \frac{\sqrt{\rho}(1-\rho^{t/2})}{1-\sqrt{\rho}}. \tag{30}$$

Consequently,

$$\sum_{p=0}^{t-1}\rho^{(t-p)/2}\sum_{i=1}^{N}\sum_{k=0}^{K-1}\|\nabla F_i(x_i^{p,k},\xi_i^{p,k})\|_2^2$$
$$\le 2KNA^2W_t + 2B^2\sum_{p=0}^{t-1}\rho^{(t-p)/2}\sum_{i=1}^{N}\sum_{k=0}^{K-1}\|\nabla f_\mathcal{I}(x_i^{p,k})\|_2^2$$
$$+ 2\sum_{p=0}^{t-1}\rho^{(t-p)/2}\sum_{i=1}^{N}\sum_{k=0}^{K-1}\|z_i^{p,k}\|_2^2. \tag{31}$$

Since $\rho^{(t-p)/2} \le 1$, from Eq. (27), with probability at least $1-\beta/2$, we have

$$\sum_{p=0}^{t-1}\rho^{(t-p)/2}\sum_{i=1}^{N}\sum_{k=0}^{K-1}\|\nabla f_\mathcal{I}(x_i^{p,k})\|_2^2 \le NH\,\mathcal{R}_H(\beta/2). \tag{32}$$

Next, we control the noise term. Let $M_t := NKt$ be the total number of stochastic gradients appearing in the bound. Since we assume that $\mathbb{E}\left[\exp\left(\frac{\|z_i^{p,k}\|_2^2}{\sigma_i^2}\right) \mid \mathcal{F}^{p,k}\right] \le \exp(1) = e$.

Let $\varsigma = \sigma_i^2\left(1 + \log\frac{2M_t}{\beta}\right)$. By the light-tailed noise condition and Markov's inequality, we have

$$\mathbb{P}\left(\|z_i^{p,k}\|_2^2 > \varsigma \mid \mathcal{F}^{p,k}\right) = \mathbb{P}\left(\exp\left(\frac{\|z_i^{p,k}\|_2^2}{\sigma_i^2}\right) > \exp\left(\frac{\varsigma}{\sigma_i^2}\right) \mid \mathcal{F}^{p,k}\right)$$
$$\le \frac{\mathbb{E}\left[\exp\left(\frac{\|z_i^{p,k}\|_2^2}{\sigma_i^2}\right) \mid \mathcal{F}^{p,k}\right]}{\exp\left(1 + \log\frac{2M_t}{\beta}\right)}$$

$$\leq \frac{e}{e \cdot \frac{2M_t}{\beta}} = \frac{\beta}{2M_t}. \tag{33}$$

That is,

$$\mathbb{P}\left( \|z_i^{p,k}\|_2^2 > \sigma_i^2 \left( 1 + \log \frac{2M_t}{\beta} \right) \Big| \mathcal{F}_{p,k} \right) \leq \frac{\beta}{2M_t}. \tag{34}$$

Taking a union bound over all $(i, p, k)$ yields that, with probability at least $1 - \beta/2$,

$$\|z_i^{p,k}\|_2^2 \leq \sigma_*^2 \left( 1 + \log \frac{2NH}{\beta} \right) \tag{35}$$

simultaneously for all $i, p, k$.

Hence, on this event,

$$\sum_{p=0}^{t-1} \rho^{(t-p)/2} \sum_{i=1}^{N} \sum_{k=0}^{K-1} \|z_i^{p,k}\|_2^2 \leq NH\sigma_*^2 \left( 1 + \log \frac{2NH}{\beta} \right). \tag{36}$$

Combining the two high-probability events by a union bound, with probability at least $1 - \beta$,

$$\sum_{p=0}^{t-1} \rho^{(t-p)/2} \sum_{i=1}^{N} \sum_{k=0}^{K-1} \|\nabla F_i(x_i^{p,k}, \xi_i^{p,k})\|_2^2$$

$$\leq 2KNA^2W_t + 2B^2NH\,\mathcal{R}_H(\beta/2) + 2NH\sigma_*^2 \left( 1 + \log \frac{2NH}{\beta} \right). \tag{37}$$

For any $\beta \in (0, 1)$, we define

$$\mathcal{B}_\Omega(t, \beta) := \frac{4\gamma^2 K \sqrt{\rho}}{1 - \sqrt{\rho}} \left[ 2KNA^2W_t + 2B^2NH\,\mathcal{R}_H(\beta/2) + 2NH\sigma_*^2 \left( 1 + \log \frac{2NH}{\beta} \right) \right]. \tag{38}$$

Then we have

$$\|x_i^t - \bar{x}^t\|_2^2 \leq \mathcal{B}_\Omega(t, \beta). \tag{39}$$

Finally, for the fixed client $c$,

$$\Omega_c(\mathcal{I}, \bar{x}^t) = \frac{1}{N-1} \|x_c^t - \bar{x}^t\|_2 \leq \frac{1}{N-1} \sqrt{\mathcal{B}_\Omega(t, \beta)}. \tag{40}$$

This completes the proof. $\qquad\square$

### A.1.2. PROOF OF THEOREM 4.4

*Proof.* Let $u_i = \frac{1}{N}$ denote the weight of peer-to-peer influence of each client before the unlearning process and $\nu_i = \frac{1}{N-1} \cdot (1 - \mathbb{1}_c(i))$ denote the updated weight of each client (including client-$c$) after client-$c$ requests deletion. Then we have

$$\|\bar{\omega}^{t+1} - \bar{x}^{t+1}\| = \|\sum_{i=1}^{N} \nu_i \omega_i^{t+1} - \sum_{i=1}^{N} u_i x_i^{t+1}\|$$

$$= \|\sum_{i=1}^{N} \nu_i (\omega_i^{t+1} - x_i^{t+1}) + \sum_{i=1}^{N} (\nu_i - u_i) x_i^{t+1}\|$$

$$= \|\sum_{i=1}^{N} \nu_i (\omega_i^{t+1} - x_i^{t+1}) + \sum_{i \neq c} \frac{1}{N-1} (x_i^{t+1} - \bar{x}^{t+1}) - \sum_{i=1}^{N} \frac{1}{N} (x_i^{t+1} - \bar{x}^{t+1})\|$$

$$\leq \|\sum_{i=1}^{N}\nu_i(\omega_i^{t+1}-x_i^{t+1})\| + \|\sum_{i\neq c}\frac{1}{N-1}(x_i^{t+1}-\bar{x}^{t+1}) - \sum_{i=1}^{N}\frac{1}{N}(x_i^{t+1}-\bar{x}^{t+1})\|$$

$$\leq \|\sum_{i=1}^{N}\nu_i(\omega_i^{t+1}-x_i^{t+1})\| + \|\frac{1}{N-1}(\sum_{i=1}^{N}x_i^{t+1}-x_c^{t+1}) - \frac{1}{N}\sum_{i=1}^{N}x_i^{t+1}\|$$

$$\leq \|\sum_{i=1}^{N}\nu_i(\omega_i^{t+1}-x_i^{t+1})\| + \|(\frac{1}{N-1}-\frac{1}{N})\sum_{i=1}^{N}x_i^{t+1} - \frac{1}{N-1}x_c^{t+1}\|$$

$$= \|\sum_{i=1}^{N}\nu_i(\omega_i^{t+1}-x_i^{t+1})\| + \frac{1}{N-1}\|\frac{1}{N}\sum_{i=1}^{N}x_i^{t+1} - x_c^{t+1}\|$$

$$= \|\sum_{i=1}^{N}\nu_i(\omega_i^{t+1}-x_i^{t+1})\| + \frac{1}{N-1}\|\bar{x}^{t+1} - x_c^{t+1}\|$$

$$\leq \sum_{i=1}^{N}\nu_i\|\omega_i^{t+1}-x_i^{t+1}\| + \Omega_c(\mathcal{I},\bar{x}^{t+1})$$

$$\leq \max_{i\in\mathcal{I}}\|\omega_i^{t+1}-x_i^{t+1}\| + \Omega_c(\mathcal{I},\bar{x}^{t+1}) \tag{41}$$

Now, for any $i \in \mathcal{I}$, let us give an upper bound for $\|\omega_i^{t+1}-x_i^{t+1}\|$ with a generic function $G$. The specific results in the 3 different cases will then be derived directly by specifying $G$ depending on the hypothesis (Hardt et al., 2016).

First, we define the one SGD step as $\mathcal{G}(f_i,\gamma,x_i^{t,k}) = x_i^{t,k} - \gamma\nabla F_i(x_i^{t,k},\xi_i^{t,k})$. Leveraging the contractivity of one step gradient descent for the local update of client-$i$, we have:

$$\|\omega_i^{t,k+1}-x_i^{t,k+1}\| = \|\mathcal{G}(f_i,\gamma,\omega_i^{t,k})-\mathcal{G}(f_i,\gamma,x_i^{t,k})\|$$
$$\leq G(f_i,\gamma)\cdot\|\omega_i^{t,k}-x_i^{t,k}\| \tag{42}$$

By applying this equation recursively $K$ times, we get

$$\|\omega_i^{t,K}-x_i^{t,K}\| \leq G(f_i,\gamma)^K\cdot\|\omega_i^t-x_i^t\| \tag{43}$$

Depending on the assumptions made on $f$, we can get 3 distinct forms of $G(f_i,\gamma)$ with their respective assumptions. Let $G(f_\mathcal{I},\gamma)$ denote $G(f_i,\gamma)$ for every $i \in \mathcal{I}$. The following results come from the stability of stochastic gradient descent, which have been discussed in (Hardt et al., 2016) (Lemma 3.7). We utilize this result as follows:

1. If $f_i$ is $L$-smooth, then $G(f_\mathcal{I},\gamma) = 1 + \gamma\cdot L$.

2. If $f_i$ is $L$-smooth, convex and $\gamma \leq \frac{2}{L}$, then $G(f_\mathcal{I},\gamma) = 1$.

3. If $f_i$ is $L$-smooth, $\lambda$-strongly convex and $\gamma \leq \frac{2}{L+\lambda}$, then $G(f_\mathcal{I},\gamma) = 1 - \frac{\gamma L\lambda}{L+\lambda}$.

Next, based on the forms of $\omega_i^{t+1}$ and $x_i^{t+1}$, we can obtain:

$$\|\omega_i^{t+1}-x_i^{t+1}\| = \|\sum_{j\in\mathcal{I}_c}\mathbf{Q}_{ij}^{t}{}'\omega_j^{t,K} - \sum_{j\in\mathcal{I}}\mathbf{Q}_{ij}^{t}x_j^{t,K}\|$$
$$\leq \max_{j}\|\omega_j^{t,K}-x_j^{t,K}\|$$
$$\leq G(f_\mathcal{I},\gamma)^K\cdot\max_{j}\|\omega_j^t-x_j^t\| \tag{44}$$

Now we get:

$$\|\bar{\omega}^{t+1}-\bar{x}^{t+1}\| \leq G(f_\mathcal{I},\gamma)^K\cdot\max_{j}\|\omega_j^t-x_j^t\| + \Omega_c(\mathcal{I},\bar{x}^{t+1}) \tag{45}$$

Finally, since $\omega_j^0 = x_j^0$ at the initial time, we can have the following bound via recurrence:

$$\|\bar{\omega}^t - \bar{x}^t\| \leq \sum_{p=0}^{t-1} G(f_{\mathcal{I}}, \gamma)^{(t-p-1)K} \cdot \Omega_c(\mathcal{I}, \bar{x}^p) \tag{46}$$

$\square$

### A.1.3. PROOF OF THEOREM 4.5

*Proof.* We denote **TRACE-DU** (Algorithm 1) as unlearning algorithm $\mathcal{M}$ in Definition 3.3 and our learning algorithm $\mathcal{A}$ follows the standard decentralized SGD (Lian et al., 2017; Koloskova et al., 2020).

Then, similar to previous studies (Sekhari et al., 2021; Suriyakumar and Wilson, 2022; Fraboni et al., 2024; Qiao et al., 2025a; Wu et al., 2026), we follow the same proof as in (Dwork and Roth, 2014) (Theorem A.1). Specifically, note that

$$\sigma(t, c) = \frac{\Upsilon(t, c)}{\epsilon} \cdot \sqrt{2 \ln(1.25/\delta)} \tag{47}$$

and in Theorem 4.4 we have proved the model sensitivity

$$\mathcal{S}(t, c) \leq \Upsilon(t, c) = \sum_{p=0}^{t-1} G(f_{\mathcal{I}}, \gamma)^{(t-p-1)K} \cdot \Omega_c(\mathcal{I}, \bar{x}^p). \tag{48}$$

Thus, we get the following inequality with the calibrated Gaussian noise scaled to $\mathcal{N}(0, \sigma(t, c)^2 \mathbb{I}_d)$:

$$\mathbb{P}\left(\bar{\omega}^t \in W\right) \leq e^{\epsilon} \mathbb{P}\left(\tilde{x} \in W\right) + \delta, \tag{49}$$
$$\mathbb{P}\left(\tilde{x} \in W\right) \leq e^{\epsilon} \mathbb{P}\left(\bar{\omega}^t \in W\right) + \delta. \tag{50}$$

This concludes that the unlearning process without retraining and learning algorithm satisfy $(\epsilon, \delta)$-unlearning guarantee. Finally, we further clarify that the additional retraining process in Algorithm 1 does not hurt this guarantee. It states that the composition of a mapping $m$ with an $(\epsilon, \delta)$-unlearning algorithm is also $(\epsilon, \delta)$-unlearning. Specifically, we denote the $(\epsilon, \delta)$-unlearning algorithm without retraining as $\mathcal{M}^- : \mathcal{P} \times \mathcal{Z} \to \mathcal{W}$ and denote $m : \mathcal{W} \to \mathcal{W}'$ be a randomized mapping (e.g., the retraining process). Then we let $m$ be a deterministic function, fix any pair of neighboring sets $a, b$ with $\|a - b\|_1 \leq 1$, and fix any event $\mathcal{X} \subseteq \mathcal{W}'$. Let $\mathcal{T} = \{r \in \mathcal{W} : m(r) \in \mathcal{X}\}$. We then have:

$$\begin{aligned}
\mathbb{P}\left[m(\mathcal{M}^-(a)) \in \mathcal{X}\right] &= \mathbb{P}\left[\mathcal{M}^-(a) \in \mathcal{T}\right] \\
&\leq e^{\epsilon} \mathbb{P}\left[\mathcal{M}^-(b) \in \mathcal{T}\right] + \delta \\
&= e^{\epsilon} \mathbb{P}\left[m(\mathcal{M}^-(b)) \in \mathcal{X}\right] + \delta.
\end{aligned} \tag{51}$$

This result proves that $m \circ \mathcal{M}^-$ (i.e., Algorithm 1) satisfies $(\epsilon, \delta)$-unlearning with the learning algorithm and aligns with that differential privacy is immune to post-processing (Dwork and Roth, 2014). $\square$

### A.1.4. PROOF OF COROLLARY 4.6

*Proof.* We denote **TRACE-DU** as the unlearning algorithm $\mathcal{M}$ in Definition 3.3, and denote by $\mathcal{A}$ the original decentralized learning algorithm. Let $U$ be the checkpoint selected for perturbation. We first prove the guarantee for the perturbation step without the subsequent retraining process, and then use post-processing to complete the proof.

Recall from Theorem 4.4 that the model sensitivity with respect to client-$c$ satisfies

$$\mathcal{S}(U, c) = \|\bar{\omega}^U - \bar{x}^U\|_2 \leq \sum_{p=0}^{U-1} G(f_{\mathcal{I}}, \gamma)^{(U-p-1)K} \cdot \Omega_c(\mathcal{I}, \bar{x}^p), \tag{52}$$

where $\bar{x}^U$ denotes the consensus model obtained by training on all clients in $\mathcal{I}$, and $\bar{\omega}^U$ denotes the corresponding consensus model obtained over the retained client set $\mathcal{I}_c = \mathcal{I} \setminus \{c\}$.

We now replace the trajectory-dependent term $\Omega_c(\mathcal{I}, \bar{x}^p)$ by its high-probability bound. For each $p \in \{0, \ldots, U-1\}$, choose a confidence parameter $\beta_p \in (0, 1)$ such that

$$\sum_{p=0}^{U-1} \beta_p \leq \beta. \tag{53}$$

By Lemma 4.3, for each fixed $p$ and client-$c$, we have

$$\mathbb{P}\left(\Omega_c(\mathcal{I}, \bar{x}^p) \leq \frac{1}{N-1}\sqrt{\mathcal{B}_\Omega(p, \beta_p)}\right) \geq 1 - \beta_p. \tag{54}$$

When $p = 0$, since all clients are initialized from the same model, i.e., $x_i^0 = \bar{x}^0$, we have $\Omega_c(\mathcal{I}, \bar{x}^0) = 0$. Thus we use the convention $\mathcal{B}_\Omega(0, \beta_0) = 0$.

Define the good event

$$\mathcal{E} := \left\{\Omega_c(\mathcal{I}, \bar{x}^p) \leq \frac{1}{N-1}\sqrt{\mathcal{B}_\Omega(p, \beta_p)} \text{ for all } p = 0, \ldots, U-1\right\}. \tag{55}$$

By the union bound and Eq. (53), we obtain

$$\mathbb{P}(\mathcal{E}) \geq 1 - \sum_{p=0}^{U-1} \beta_p \geq 1 - \beta. \tag{56}$$

On the event $\mathcal{E}$, substituting Eq. (55) into Eq. (52) yields

$$\mathcal{S}(U, c) = \|\bar{\omega}^U - \bar{x}^U\|_2 \leq \widehat{\Upsilon}(U, c; \beta), \tag{57}$$

where

$$\widehat{\Upsilon}(U, c; \beta) := \sum_{p=0}^{U-1} G(f_\mathcal{I}, \gamma)^{(U-p-1)K} \cdot \frac{1}{N-1}\sqrt{\mathcal{B}_\Omega(p, \beta_p)}. \tag{58}$$

We then calibrate the Gaussian perturbation according to this high-probability sensitivity bound:

$$\sigma(U, c; \beta) = \frac{\widehat{\Upsilon}(U, c; \beta)}{\epsilon}\sqrt{2\ln\frac{1.25}{\delta_\mathrm{G}}}. \tag{59}$$

Let $\mathcal{M}^-$ denote the algorithm before the subsequent retraining stage. It outputs

$$\mathcal{M}^-(\bar{x}^U) = \bar{x}^U + Z, \qquad Z \sim \mathcal{N}\left(0, \sigma(U, c; \beta)^2 \mathbb{I}_d\right). \tag{60}$$

Similarly, the corresponding perturbed retained-client model can be written as

$$\bar{\omega}^U + Z, \qquad Z \sim \mathcal{N}\left(0, \sigma(U, c; \beta)^2 \mathbb{I}_d\right). \tag{61}$$

Conditioned on the good event $\mathcal{E}$, Eq. (57) shows that the $\ell_2$ distance between $\bar{x}^U$ and $\bar{\omega}^U$ is at most $\widehat{\Upsilon}(U, c; \beta)$. Therefore, by the Gaussian mechanism with $\ell_2$ sensitivity $\widehat{\Upsilon}(U, c; \beta)$ and noise scale in Eq. (59), for any measurable set $W \subseteq \mathcal{W}$,

$$\mathbb{P}\left(\bar{x}^U + Z \in W \mid \mathcal{E}\right) \leq e^\epsilon \mathbb{P}\left(\bar{\omega}^U + Z \in W \mid \mathcal{E}\right) + \delta_\mathrm{G}. \tag{62}$$

$$\mathbb{P}\left(\bar{\omega}^U + Z \in W \mid \mathcal{E}\right) \leq e^\epsilon \mathbb{P}\left(\bar{x}^U + Z \in W \mid \mathcal{E}\right) + \delta_\mathrm{G}. \tag{63}$$

We now remove the conditioning on $\mathcal{E}$. For the forward direction, we have

$$\mathbb{P}\left(\mathcal{M}^-(\bar{x}^U) \in W\right) = \mathbb{P}\left(\bar{x}^U + Z \in W, \mathcal{E}\right) + \mathbb{P}\left(\bar{x}^U + Z \in W, \mathcal{E}^c\right)$$

$$\leq \mathbb{P}(\mathcal{E}) \left[ e^\epsilon \mathbb{P}\left(\bar{\omega}^U + Z \in W \mid \mathcal{E}\right) + \delta_{\mathrm{G}} \right] + \mathbb{P}(\mathcal{E}^c)$$
$$\leq e^\epsilon \mathbb{P}\left(\bar{\omega}^U + Z \in W\right) + \delta_{\mathrm{G}} + \beta. \tag{64}$$

In the last inequality, we used $\mathbb{P}(\mathcal{E}^c) \leq \beta$ from Eq. (56). Therefore,

$$\mathbb{P}\left(\mathcal{M}^-(\bar{x}^U) \in W\right) \leq e^\epsilon \mathbb{P}\left(\bar{\omega}^U + Z \in W\right) + \delta_{\mathrm{G}} + \beta. \tag{65}$$

By the same argument with $\bar{x}^U$ and $\bar{\omega}^U$ exchanged, we also obtain

$$\mathbb{P}\left(\bar{\omega}^U + Z \in W\right) \leq e^\epsilon \mathbb{P}\left(\mathcal{M}^-(\bar{x}^U) \in W\right) + \delta_{\mathrm{G}} + \beta. \tag{66}$$

Equations (65) and (66) imply that the perturbation-only unlearning algorithm $\mathcal{M}^-$ satisfies $(\epsilon, \delta_{\mathrm{G}} + \beta)$-unlearning.

It remains to show that the subsequent retraining stage in **TRACE-DU** does not weaken the guarantee. Let $m : \mathcal{W} \to \mathcal{W}'$ denote the subsequent decentralized retraining procedure initialized from the perturbed checkpoint. This retraining procedure may be deterministic or randomized, but it only takes the perturbed model as input and does not access the deleted client's data. Thus it is a post-processing operation. For any measurable set $\mathcal{X} \subseteq \mathcal{W}'$, fix the randomness of $m$ if $m$ is randomized, and define

$$\mathcal{T} := \{r \in \mathcal{W} : m(r) \in \mathcal{X}\}. \tag{67}$$

Using the already established guarantee of $\mathcal{M}^-$, we get

$$\mathbb{P}\left(m(\mathcal{M}^-(\bar{x}^U)) \in \mathcal{X}\right) = \mathbb{P}\left(\mathcal{M}^-(\bar{x}^U) \in \mathcal{T}\right)$$
$$\leq e^\epsilon \mathbb{P}\left(\bar{\omega}^U + Z \in \mathcal{T}\right) + \delta_{\mathrm{G}} + \beta$$
$$= e^\epsilon \mathbb{P}\left(m(\bar{\omega}^U + Z) \in \mathcal{X}\right) + \delta_{\mathrm{G}} + \beta. \tag{68}$$

The reverse direction follows identically. Therefore, the full **TRACE-DU** algorithm $\mathcal{M} = m \circ \mathcal{M}^-$ satisfies $(\epsilon, \delta_{\mathrm{G}} + \beta)$-unlearning.

Finally, if we set

$$\delta_{\mathrm{G}} = \frac{\delta}{2}, \qquad \beta = \frac{\delta}{2}, \tag{69}$$

then $\delta_{\mathrm{G}} + \beta = \delta$. In this case, **TRACE-DU** satisfies $(\epsilon, \delta)$-unlearning with the noise scale

$$\sigma(U, c; \delta/2) = \frac{\widehat{\Upsilon}(U, c; \delta/2)}{\epsilon} \sqrt{2 \ln \frac{2.5}{\delta}}. \tag{70}$$

This completes the proof. □

### A.1.5. PROOF OF THEOREM 4.7

*Proof.* For sequential unlearning requests, we begin with $u = 1$ (i.e., the first unlearning request $\mathcal{Q}_1$). By Theorem 4.5, perturbing the checkpoint $\bar{x}_0^{U_1}$ yields the first perturbed model $\bar{x}_1^0$, which satisfies the desired client-wise unlearning guarantee for $\mathcal{Q}_1$.

First we consider the simpler case: $\forall u$, $U_u \leq U_{u+1}$. It implies that the model $H_u(U_{u+1})$ that requires adding noise at $(u + 1)$-th unlearning appears later in the training history than previous perturbed model $\bar{x}_u^0$ (which is obtained by adding noise to $H_{u-1}(U_u)$). We assume that $(\epsilon, \delta)$-unlearning guarantees are achieved for every client in $\mathcal{V}_u$. Then it is obvious that the next perturbed model $\bar{x}_{u+1}^0$ guarantees $(\epsilon, \delta)$-unlearning of every client in $\mathcal{V}_{u+1}$.

Next, we consider another case: $\exists$ unlearning request $u' \leq u$ such that $U_{u+1} < U_{u'}$. It implies that the model $H_u(U_{u+1})$ requires adding noise at $(u + 1)$-th unlearning request appears earlier than previous perturbed model $\bar{x}_{u'}^0$ (which is obtained by adding noise to $H_{u'-1}(U_{u'})$). Thus we have $\Upsilon_{u'}(U_{u+1}, \mathcal{Q}_{u'}) \leq \Upsilon_{u'}(U_{u'}, \mathcal{Q}_{u'})$. This implies that perturbing the model $H_u(U_{u+1})$ with noise $\mathcal{N}(0, \sigma(U_{u'}, \mathcal{Q}_{u'}\mathbb{I}_d)$ guarantees $(\epsilon, \delta)$-unlearning of the clients in $\mathcal{Q}_{u'}$. We extend this to all unlearning requests $u$ such that $U_{u+1} < U_u$ and can conclude that the next perturbed model $\bar{x}_{u+1}^0$ guarantees $(\epsilon, \delta)$-unlearning of every client in $\mathcal{V}_{u+1}$ by induction.

For the high-probability guarantee, the argument follows the same induction structure, with the deterministic sensitivity bound replaced by the high-probability sensitivity bound. Specifically, for the $(u+1)$-th unlearning request $\mathcal{Q}_{u+1}$, let $\{\beta_{u+1,p}\}_{p=0}^{U_{u+1}-1}$ satisfy $\sum_{p=0}^{U_{u+1}-1} \beta_{u+1,p} \le \beta_{u+1}$. For each client $c \in \mathcal{Q}_{u+1}$, define

$$\widehat{\Upsilon}_{u+1}(U_{u+1}, c; \beta_{u+1}) := \sum_{p=0}^{U_{u+1}-1} G(f_{\mathcal{I}}, \gamma)^{(U_{u+1}-p-1)K} \cdot \frac{1}{N-1} \sqrt{B_\Omega(p, \beta_{u+1,p})}. \tag{71}$$

For the whole request set, we take the worst-case bound

$$\widehat{\Upsilon}_{u+1}(U_{u+1}, \mathcal{Q}_{u+1}; \beta_{u+1}) := \max_{c \in \mathcal{Q}_{u+1}} \widehat{\Upsilon}_{u+1}(U_{u+1}, c; \beta_{u+1}). \tag{72}$$

By Lemma 4.3 and the union bound over $p = 0, \ldots, U_{u+1} - 1$, with probability at least $1 - \beta_{u+1}$, the sensitivity of every client $c \in \mathcal{Q}_{u+1}$ at the selected checkpoint is bounded by

$$\mathcal{S}_{u+1}(U_{u+1}, c) \le \widehat{\Upsilon}_{u+1}(U_{u+1}, c; \beta_{u+1}) \le \widehat{\Upsilon}_{u+1}(U_{u+1}, \mathcal{Q}_{u+1}; \beta_{u+1}). \tag{73}$$

Therefore, if **TRACE-SDU** perturbs $H_u(U_{u+1})$ with Gaussian noise $\mathcal{N}\left(0, \sigma(U_{u+1}, \mathcal{Q}_{u+1}; \beta_{u+1})^2 \mathbb{I}_d\right)$, where

$$\sigma(U_{u+1}, \mathcal{Q}_{u+1}; \beta_{u+1}) = \frac{\widehat{\Upsilon}_{u+1}(U_{u+1}, \mathcal{Q}_{u+1}; \beta_{u+1})}{\epsilon} \sqrt{2 \ln \frac{1.25}{\delta_G^{u+1}}}. \tag{74}$$

Then, conditioned on the above good event, the Gaussian mechanism gives $(\epsilon, \delta_G^{u+1})$-unlearning for clients in $\mathcal{Q}_{u+1}$. With an additional failure probability $\beta_{u+1}$, the $(u+1)$-th perturbation satisfies $(\epsilon, \delta_G^{u+1} + \beta_{u+1})$-unlearning.

Then we verify that this additional high-probability perturbation does not invalidate the guarantees of previously deleted clients. This follows from the same two case argument above. If the newly selected checkpoint $U_{u+1}$ is later than a previous checkpoint $U_{u'}$, then all operations after the previous perturbation only use retained clients and are therefore post-processing. If $U_{u+1} < U_{u'}$ for some previous request $u'$, then the newly selected earlier checkpoint also satisfies the checkpoint selection condition for the corresponding historical sensitivity bound, so the newly injected noise is sufficient to cover the previous request as well. Thus, by induction, after the $(u+1)$-th request, all clients in $\mathcal{V}_{u+1}$ remain protected.

Finally, the subsequent decentralized training initialized from the perturbed model $\bar{x}_{u+1}^0$ only uses the retained client set $\mathcal{I}_{u+1}$, and hence is a post-processing operation. Consequently, it does not weaken the above high-probability unlearning guarantee. In a word, after each sequential unlearning request, the currently released model produced by **TRACE-SDU** satisfies client-wise $(\epsilon, \delta)$-unlearning with respect to the clients deleted so far, under the corresponding per-request noise calibration. This completes the proof of the high-probability case.

$\square$

## A.2. Missing Details in Experiments

### A.2.1. IMPLEMENTATION DETAILS

For the sake of experimental fairness, we maintain consistent settings for the hyperparameters listed below across all evaluated scenarios and methods.

To better simulate practical scenarios, we evaluate all methods under non-IID data distributions following a Dirichlet distribution with $\alpha = 0.3$.

Each client trains with a batch size of 64 for 2 epochs per round. Consistent with the configurations reported in their original papers (Qiao et al., 2025a; Wu et al., 2026), we set the learning rate to 0.001 for all baselines, ensuring they achieve optimal performance. For the unlearning budget, we set $\epsilon = 50$ and $\delta = 10^{-5}$. We also discuss the algorithmic robustness under different noise scales from 0.025 to 0.2 in the following part.

### A.2.2. BASELINE METHODS

In our experiments, we compare our framework against the following baseline DU methods:

**Retrain (D-PSGD)** (Lian et al., 2017) refers to the process of retraining from scratch using standard decentralized SGD solely on retained clients when specific clients issue unlearning requests. This approach represents exact unlearning and serves as the gold standard for verifying unlearning performance.

**PDUDT** (Qiao et al., 2025a) was the first provable decentralized unlearning algorithm with certified unlearning guarantee. This method utilizes cached statistics of each client such as gradients and intermediate model parameters to compute the approximations of the gradient residuals and add related noise perturbation. Its outstanding unlearning performance was verified by extensive experiments.

**CDU** (Wu et al., 2026) introduced a certified decentralized unlearning algorithm leveraging Newton-style model update. When a client requests deletion, it first performs model correction with curvature information, then the updated model is further perturbed with Gaussian noise and broadcast to retained clients. It is quite efficient due to the effective Newton-style model correction. Although the theoretical results in their paper heavily rely on convex loss settings, the authors empirically demonstrated its performance for non-convex tasks.

### A.2.3. EVALUATION METRICS

**Model Utility** We utilize retain accuracy to evaluate model utility. **Retain Accuracy** represents the test accuracy on the retained clients and serves as a quantitative measure of model utility. Specifically, retain accuracy indicates how well the unlearned models preserve its performance relative to the original state (i.e., before unlearning). Therefore, higher values are desirable for this metric.

**Unlearning Quality** We utilize forget accuracy to evaluate unlearning quality. **Forget Accuracy** is the test accuracy on the unlearned clients. Since a drop in forget accuracy implies the effective removal of the specific clients' influence, it should be as low as possible. Instead of adopting the simplest comparison, we utilize a different approach to compare forget accuracy. Specifically, we can calculate the gap compared to *RT*, with a smaller gap implying more effective unlearning. This approach has been adopted by prior work (Fraboni et al., 2024; Khalil et al., 2025), confirming its fairness and effectiveness. Additionally, the unlearning effectiveness could also be evaluated by addressing backdoor attacks (Khalil et al., 2025). In decentralized systems where devices are inherently resource-constrained, leveraging computationally efficient backdoor defenses (Yuan et al., 2025; Xu et al., 2025) could effectively enhance the overall unlearning quality.

**Privacy Protection** We utilize MIA accuracy to evaluate the privacy protection performance of unlearning baselines. **Membership Inference Attack (MIA)** aims to determine whether specific data samples were used during model training. For effective unlearning, it is imperative to prevent attackers from discerning whether data from unlearned clients participated in the training process. Specifically, MIA success rates approaching 50% imply that the attack performs no better than random guessing, indicating an inability to reliably identify training membership. Such results effectively demonstrate effective data removal. Therefore, we utilize MIA on the unlearned model to verify whether the algorithm has successfully eliminated the unlearned clients' influence. It serves as a dual metric for privacy protection performance and unlearning quality.

**Unlearning Efficiency** We utilize unlearning time to evaluate unlearning efficiency. **Unlearning Time** denotes the runtime required to execute the unlearning operations. As the subsequent training phase employs an identical number of rounds for all algorithms, its runtime is excluded from the measurement, in alignment with prior works (Qiao et al., 2025a; Wu et al., 2026). For our proposed method, the unlearning time comprises checkpoint selection and noise perturbation. For *PDUDT*, it accounts for the computation of gradient residuals, weighting factors, and noise addition. For *CDU*, it encompasses the time for model correction, noise injection, and broadcasting.

### A.2.4. UNLEARNING PERFORMANCE

***Single-DU.*** We first discuss the results in *Single-DU* scenario. Table 6 reports the optimal retain accuracy achieved by various methods across different datasets under the *Single-DU* scenario. In Table 7, we present the absolute difference between the accuracy of each method and *Retrain* baseline. This evaluation metric aligns with established protocols in prior work (Khalil et al., 2025).

***Seq-DU.*** Next, we provide more details and address potential points of confusion regarding Figure 2. First, consistent with prior studies (Qiao et al., 2025a; Wu et al., 2026), we standardize the learning algorithm as D-PSGD (Lian et al., 2017)

across all methods. Under identical hyperparameter settings, we conduct a pre-training phase of 40 rounds before the issue of the first unlearning request. At this stage, all methods achieve the same initial accuracy, denoted as unlearning request index 0 (where the retained client set $\mathcal{I}_0 = \mathcal{I}$). To be specific, the initial accuracy is $79.86\%$ on MNIST and $65.01\%$ on Fashion-MNIST, corresponding to the points at request index 0 in Figure 2.

*Table 6.* Comparison of best retain accuracy on different datasets under *Single-DU*.

| METHOD | BEST RETAIN ACCURACY (%) | | |
|---|---|---|---|
| | MNIST | F-MNIST | CIFAR-10 |
| RETRAIN | 87.89 | 82.41 | 75.95 |
| PDUDT | 84.80 | 84.53 | 90.68 |
| CDU | 86.86 | 85.09 | 92.29 |
| **TRACE-DU** | **88.09** | **89.12** | **95.95** |

*Table 7.* Comparison of forget accuracy gap on different datasets under *Single-DU*.

| METHOD | FORGET GAP $\Delta$ (%) | | |
|---|---|---|---|
| | MNIST | F-MNIST | CIFAR-10 |
| RETRAIN | 0.00 | 0.00 | 0.00 |
| PDUDT | 1.25 | 4.61 | 31.93 |
| CDU | 6.12 | 2.35 | **13.33** |
| **TRACE-DU** | **0.35** | **0.67** | 21.75 |

Furthermore, it is necessary to clarify the pronounced sharp increase in retain accuracy observed between request indices 0 and 1 in Figure 2. Specifically, the reported accuracy for each request is evaluated on the retained clients after executing the unlearning algorithm and completing 40 rounds of subsequent training. At index 0, the model has undergone only 40 initial training rounds, which is insufficient to reach optimal accuracy. However, upon the first unlearning request (index 1), the model receives 40 extra rounds of subsequent training beyond the unlearning process, allowing it to approach convergence. Although a small fraction of clients depart (1 or 3 out of 50 clients), the minor performance degradation caused by their exit is vastly outweighed by the substantial accuracy gains achieved through the additional optimization rounds.

**Explanation for the Inferior Performance of *Retrain* in *Seq-DU*.** To begin with, we argue that the constraints imposed by prior works are impractical for real-world applications. *CDU* requires model convergence before unlearning (Wu et al., 2026), while *PDUDT* necessitates an excessively long subsequent training phase ($T_r = 200$) (Qiao et al., 2025a).

Aiming to reflect real-world dynamics, our setting allows clients to initiate unlearning at any training stage and supports high-frequency unlearning requests. Specifically, the initial unlearning request is scheduled after 40 rounds of training. Thereafter, the interval between subsequent unlearning requests is also fixed at 40 rounds.

While the initial performance boost (Index 0 to 1) has been clarified, we proceed to explain the subsequent performance decline of *Retrain* as sequential unlearning requests accumulate. In non-IID settings, the cumulative withdrawal of clients exacerbates data scarcity, rendering the remaining dataset less representative. Consequently, the distribution learned by the model diverges increasingly from the true global distribution, leading to a decline in retain accuracy. Furthermore, for each unlearning request, the *Retrain* method necessitates training from scratch on retained clients. However, the frequent withdrawal of clients makes it challenging to fully restore optimal accuracy within the limited interval between requests. Consequently, the combined effect of distribution shifts and frequent unlearning makes the performance of *Retrain* method unsatisfactory in *Seq-DU* scenarios.

### A.3. Additional Experimental Results

In this section, we provide supplementary experimental results and discussion to complement the main text.

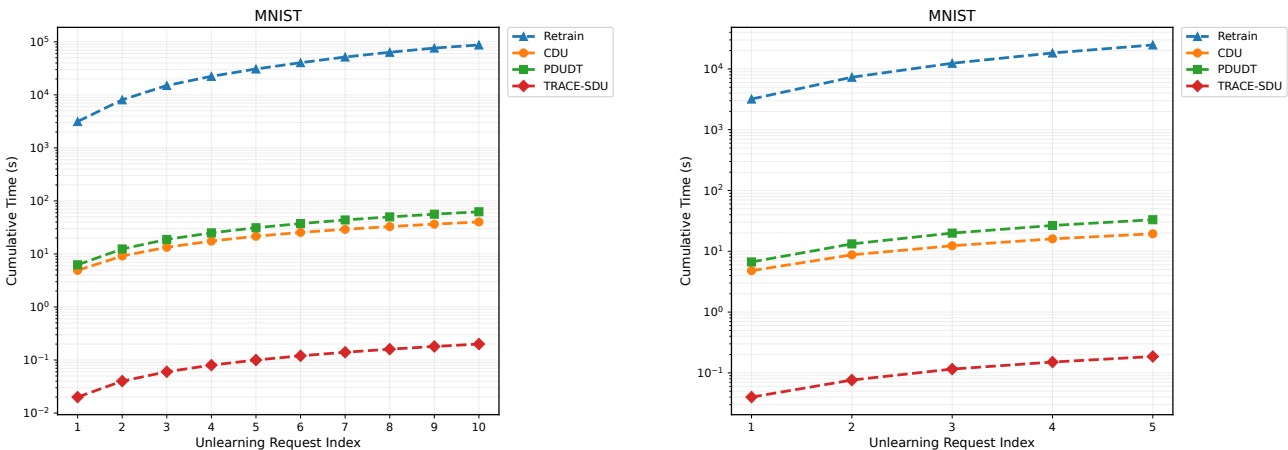

*Figure 4.* Comparison of cumulative unlearning time across different methods for sequential unlearning requests on the MNIST dataset. *Left*: 10 sequential unlearning requests, removing one client per request. *Right*: 5 sequential unlearning requests, removing 3 clients per request.

### A.3.1. UNLEARNING EFFICIENCY

Figure 4 presents the cumulative unlearning runtime results for sequential requests on the MNIST dataset.

### A.3.2. EXTRA STORAGE OVERHEAD COMPARISON

To quantify the extra storage overhead incurred during the unlearning process, we compare our method with PDUDT under identical training settings. In PDUDT, it requires each client-$i$ to save the gradients of its neighbors and the required memory increases linearly as the number of neighbors grows (Qiao et al., 2025a). Each checkpoint stores clients' gradient tensors as float32.

In contrast, our method only records consensus model snapshots obtained by weighted averaging of active clients' models, and scalar $L_2$ distances between each client's local model and the corresponding consensus model. The consensus model snapshots are stored as float32 tensors, while each scalar distance is conservatively counted as a float64 value. We report the measured storage in bytes in Table 8.

*Table 8.* Comparison of extra storage overhead.

| METHOD | EXTRA STORAGE (MB) |
|--------|--------------------|
| PDUDT  | 62.55              |
| OURS   | **1.21**           |

### A.3.3. ABLATION STUDIES

**Varying Noise Scales.** To demonstrate the robustness of our algorithm, we investigate model performance under varying noise scales. Specifically, we vary the noise parameter $\sigma$ from 0.025 to 0.200 and evaluate retain accuracy across convex and non-convex settings. Figure 5 illustrates that **TRACE-DU** maintains stable performance despite variations in $\sigma$. Even when the noise scale is amplified by a factor of eight, the retain accuracy does not suffer from significant degradation, thereby effectively preserving model utility.

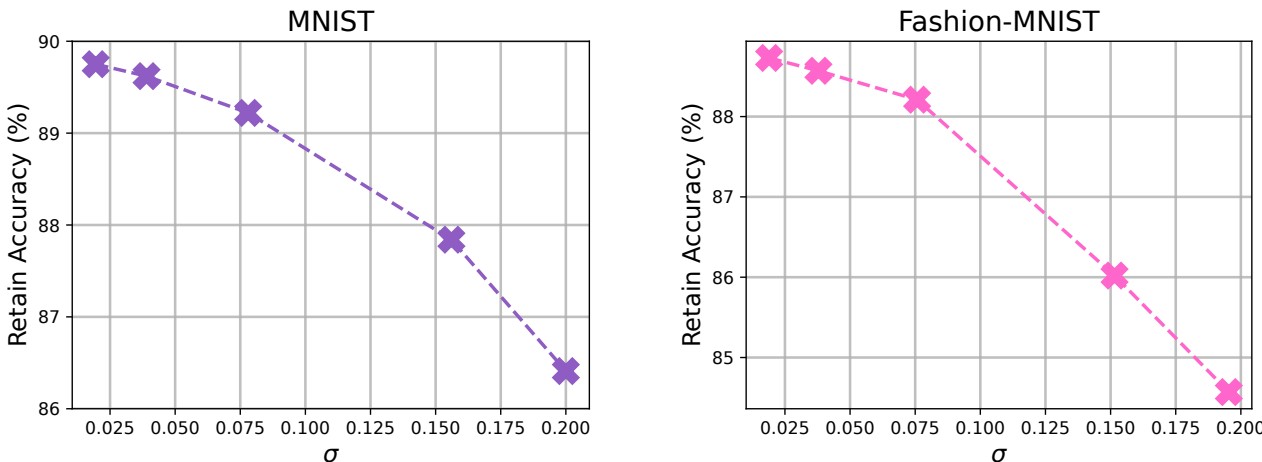

*Figure 5.* Retain accuracy of unlearned models using **TRACE-DU** under varying noise scales.

**Varying Client Scales, $\alpha$ and $\epsilon$.** The main paper reports results under non-IID data with Dirichlet $\alpha = 0.3$ and 50 clients. For the unlearning budget, we set $\epsilon = 50$ and $\delta = 10^{-5}$. We additionally evaluate a more heterogeneous setting with *70* clients and a different Dirichlet parameter $\alpha = 0.4$ under both *Single-DU* and *Seq-DU* (5 requests). For the unlearning budget, we set $\epsilon = 20$ and $\delta = 10^{-5}$. We provide the results of retain accuracy and unlearning time in Figure 6 and Figure 7, which illustrate that **TRACE-DU** and **TRACE-SDU** maintain stable performance despite variations in $\sigma$.

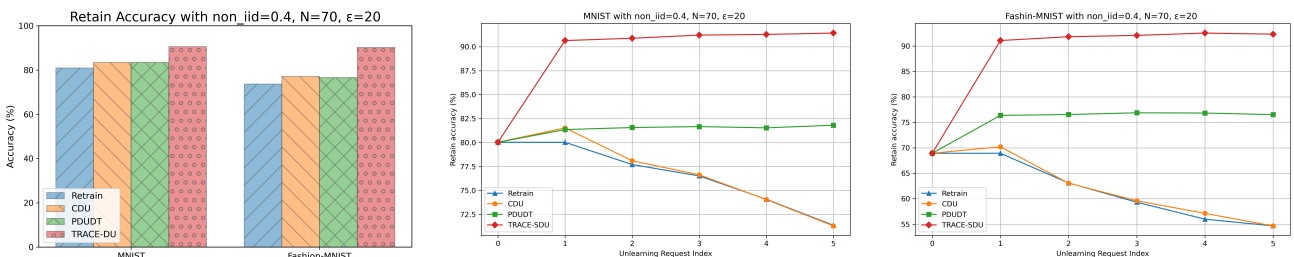

*Figure 6.* Retain accuracy of unlearned models under $N = 70$, $\alpha = 0.4$, $\epsilon = 20$.

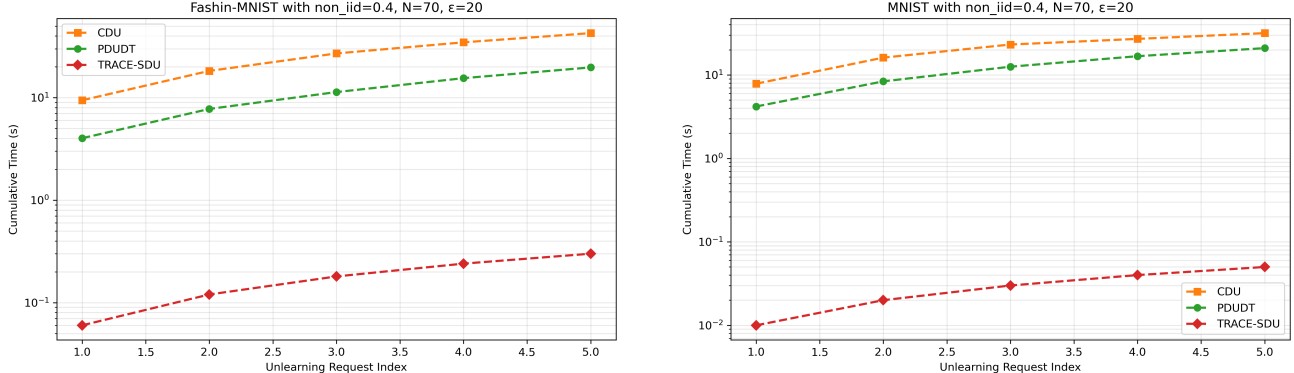

*Figure 7.* Unlearning time comparison under $N = 70$, $\alpha = 0.4$, $\epsilon = 20$.

**Varying $T_r$.** The main paper reports results with $T_r = 40$. We added ablations with $T_r = 20$ and $T_r = 60$ under both *Single-DU* and *Seq-DU* (5 requests). We provide the results of retain accuracy and unlearning time in Figure 8, Figure 9 and Figure 10.

By setting $T_r = 60$, we further demonstrate that our method retains its superiority over the baselines, even when all approaches are afforded a more generous budget for recovery rounds. Additionally, we further simulate the extremely high-frequency unlearning regime. In this regime, any method based on the perturbation and recovery mechanism may suffer from repeated injection-recovery cycles (Qiao et al., 2025a), leading to significant performance degradation and communication overhead. This is a fundamental challenge, not one unique to **TRACE-SDU**. To test the performance in high-frequency unlearning, our current *Seq-DU* setting already moves in this direction by issuing requests every 40 rounds, under which **TRACE-SDU** remains substantially more efficient than the baselines while preserving better retain utility. We further explore an extremely high-frequency unlearning setting by reducing the interval from 40 to 20 rounds. The results in Figure 8 show that **TRACE-SDU** remains stable and efficient even under more frequent requests. Finally, for communication overhead caused by high-frequency unlearning, we can utilize some compression techniques (e.g. Top-k compressor) to achieve more efficient communication. We discuss this in Appendix A.4.1.

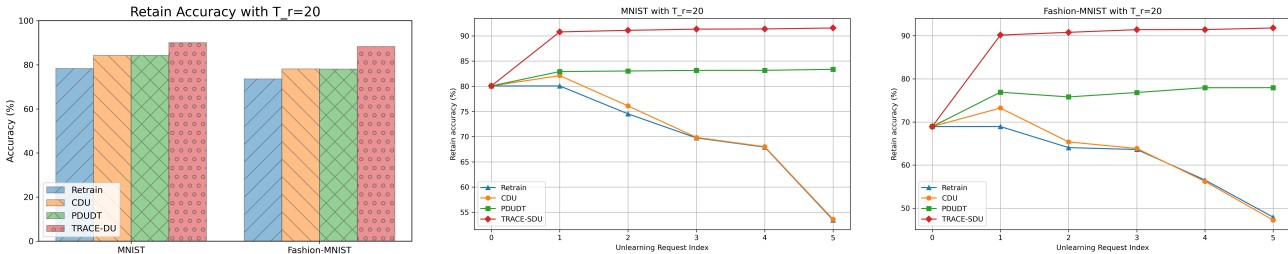

*Figure 8.* Retain accuracy of unlearned models with $T_r = 20$.

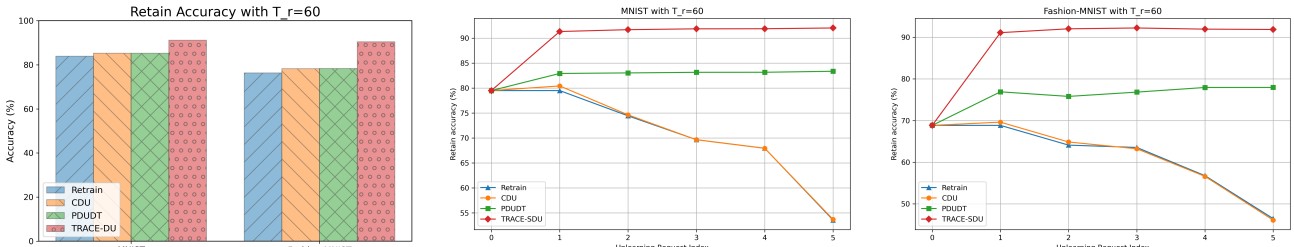

*Figure 9.* Retain accuracy of unlearned models with $T_r = 60$.

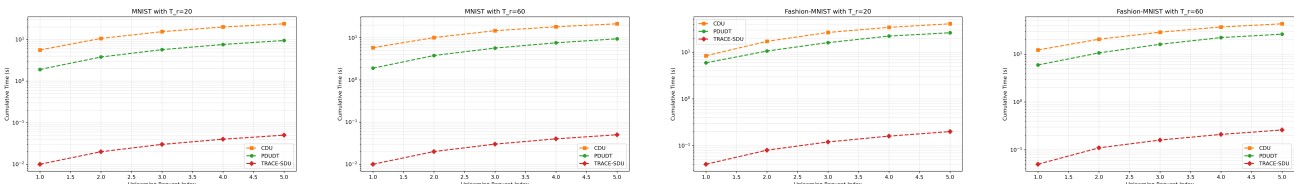

*Figure 10.* Unlearning time comparison under $T_r = 20$ and $T_r = 60$.

**Varying Topologies.**    We further report results on regular graphs with different connectivity levels under both *Single-DU* and *Seq-DU* settings. We provide results of retain accuracy and unlearning time in Figure 11 and Figure 12.

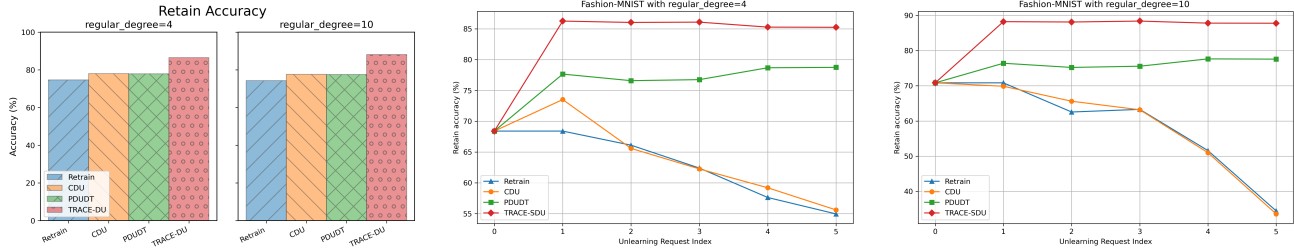

*Figure 11.* Retain accuracy of unlearned models with 4-regular graph and 10-regular graph on F-MNIST.

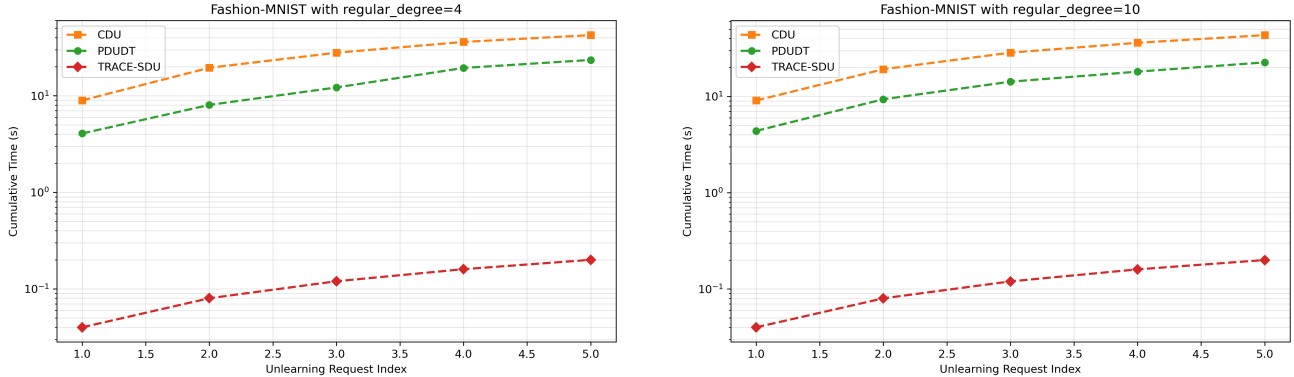

*Figure 12.* Unlearning time comparison with 4-regular graph and 10-regular graph on F-MNIST.

### A.3.4. MORE DATASETS

To further test scalability, we conducted additional experiments on SVHN dataset (Netzer et al., 2011) with 70 clients. The results are reported in Table 9.

*Table 9.* Comparison on SVHN dataset under *Single-DU* with 70 clients.

| METHOD | RETAIN ACC. (%) | TIME (S) |
|---|---|---|
| RETRAIN | 72.54 | / |
| CDU | 87.40 | 26.17 |
| **TRACE-DU** | **88.81** | **1.04** |

These results are consistent with our main findings: **TRACE-DU** preserves strong retain utility while being substantially more efficient than the baseline methods.

We further evaluated **TRACE-DU** in a text-domain setting using BERT-tiny model on AG News dataset (Zhang et al., 2015) under *Single-DU*. The results are reported in Table 10.

*Table 10.* Comparison on AG News dataset under *Single-DU* using BERT-tiny.

| METHOD | RETAIN ACC. (%) |
|---|---|
| RETRAIN | 92.02 |
| PDUDT | 92.09 |
| CDU | 87.22 |
| **TRACE-DU** | **92.62** |

The results suggest that **TRACE-DU** also generalizes well beyond CV tasks, achieving strong retain utility in the text-domain setting.

## A.4. Discussion

### A.4.1. FUTURE WORK

To further address the storage and communication overhead, we can utilize some common compression techniques such as Top-k compressor. The compression would introduce an additional aggregation error term into the sensitivity recursion. For unbiased compressors, this would typically appear as an extra controllable residual term; for biased compressors, additional control mechanisms such as error feedback or variance-bias reduction may be needed. More broadly, our framework is orthogonal to communication-efficient decentralized training: if the compressor satisfies a suitable bounded-error property, a trajectory-aware bound of similar form should still be derivable. A formal result would require specifying the compressor model and its assumptions, which we leave for future work.

### A.4.2. COMPARISON WITH PRIOR WORK

The most closely related work to ours is SIFU (Fraboni et al., 2024), which studies certified unlearning in a centralized architecture. Both SIFU and **TRACE-DU** rely on SGD stability to control the effect of removing a client. However, the two methods address fundamentally different learning protocols and require different technical tools. In SIFU, a central server directly aggregates client updates and can trace client contributions through the centralized training trajectory. In contrast, **TRACE-DU** targets fully decentralized peer-to-peer learning, where there is no central server.

This difference is not merely architectural. It changes the quantities that are observable during unlearning and the form of the sensitivity analysis. In particular, the contribution proxy used in **TRACE-DU** is derived from decentralized mixing dynamics rather than from server-side aggregation. The factor $1/(N-1)$ in our proxy could represent the asymptotically uniform influence that one client exerts on the retained clients through peer-to-peer mixing. Moreover, our sensitivity bound combines local SGD stability with the propagation of decentralized neighbor aggregation.

Notably, the high-probability sensitivity bound established in our paper further distinguishes **TRACE-DU** from SIFU. SIFU derives its certified unlearning guarantee in a centralized federated architecture, where the server can directly observe and aggregate client updates and therefore trace the contribution of a removed client along the centralized trajectory. While in fully decentralized learning, such server-side contribution tracing is not always available. The influence of a client is propagated only through repeated peer-to-peer mixing, and the exact contribution of a deleted client cannot be directly isolated from a single centralized aggregation rule. This necessitates a distinct sensitivity analysis that jointly accounts for several factors that are fundamentally absent in centralized settings, including stochastic gradient noise, consensus error and the network mixing rate.

Specifically, Lemma 4.3 shows that $\Omega_i(\mathcal{I}, \bar{x}^t)$ can be upper bounded by a quantity $\mathcal{B}_\Omega(t, \beta)$ that explicitly contains the network mixing factor $\rho$, the stochastic gradient noise level $\sigma_*^2$, the heterogeneity parameters $A$ and $B$, and the high-probability gradient energy envelope $\mathcal{R}_H(\cdot)$. In particular, the factor $\rho$ captures how fast decentralized mixing approaches ideal uniform averaging: a smaller $\rho$ leads to faster consensus and a tighter sensitivity envelope, whereas a larger $\rho$ induces a larger residual disagreement term. Thus, our bound explains how the quality of the decentralized communication graph directly affects the amount of noise required for certified unlearning. Therefore, although both methods share the broad checkpoint selection and perturbation philosophy, our framework provides a substantially different theoretical treatment: a decentralized, topology-aware, high-probability sensitivity analysis that supports certified unlearning under non-convex objectives and dynamic peer-to-peer communication.

