# OpenReview forum: "Trajectory-Aware Certified Decentralized Unlearning via SGD Stability"
_ICML.cc/2026/Conference — ICML 2026 regular_

### Official Review · Reviewer_P2MJ · 2026-03-04

**Soundness:** 3
**Presentation:** 3
**Significance:** 2
**Originality:** 3
**Overall Recommendation:** 4
**Confidence:** 2

**Summary:**

This paper proposes a method named TRACE-DU, aiming to address the challenge of machine unlearning in fully decentralized networks caused by the absence of a central server. The authors leverage the stability of local SGD updates and the dynamic trajectories of decentralized training to establish a fine-grained sensitivity analysis model. The proposed TRACE-DU can intelligently select an optimal checkpoint from the historical trajectory to inject calibrated controlled noise, followed by a small amount of recovery training. Furthermore, the method is successfully extended to handle continuous, multi-client unlearning requests.

**Compliance With Llm Reviewing Policy:**

Affirmed.

**Final Justification:**

My concerns have been addressed. I will maintain the original score.

**Key Questions For Authors:**

1. The proposed method requires the communication matrix to be strictly doubly stochastic. Is this assumption overly stringent? In scenarios where nodes drop out of the decentralized network or in truly dynamic topologies, maintaining matrix symmetry and doubly stochasticity can be quite difficult to achieve in practice. Could the authors provide experimental results showing the impact of communication matrix changes caused by node dropouts in a dynamic topology? Furthermore, does the theoretical analysis of the sensitivity derivation still hold under these conditions?

2. TRACE-SDU needs to cache the model trajectory $H_u$ locally or within the system to process future continuous unlearning requests. What is the exact storage overhead for this historical trajectory in Decentralized Federated Learning (DFL)? Could the authors provide specific quantitative results and discuss potential strategies to mitigate this memory overhead?

3. In the extreme case of high-frequency unlearning, the decentralized network might fall into a continuous loop of "noise injection followed by recovery training." This sustained communication overhead cannot be ignored in decentralized scenarios and could be considered another form of the retraining burden. Could the authors analyze the method's performance, stability, and efficiency under such extremely high-frequency scenarios?

**Limitations:**

Authors can add a limitations section to enhance the paper's rigor.

**Strengths And Weaknesses:**

Strengths:
1. Theoretically, this paper frees existing certified unlearning methods from their traditional reliance on Hessian matrices and convex loss functions. By introducing the SGD expansion factor $G(f_i, \gamma)$, the authors provide a rigorous theoretical upper bound for sensitivity, which holds significant theoretical value.
2. The paper addresses not only single unlearning requests but also extends the method to continuous, multiple unlearning scenarios. Moreover, it applies to time-varying topologies, demonstrating highly flexible applicability in practical decentralized networks.

Weaknesses:
1. The datasets used in the experiments are relatively simple. The method has not been evaluated on more complex datasets, which fails to adequately demonstrate its effectiveness and robustness under more complex data distributions.
2. After noise injection, the proposed method still requires $T_r$ rounds of subsequent training to recover model utility. Although this is significantly more efficient than complete retraining, repeatedly initiating $T_r$ rounds of decentralized communication and aggregation in high-frequency unlearning scenarios remains a non-negligible communication bottleneck.
3. To enable checkpoint selection and continuous unlearning, TRACE-SDU requires maintaining a historical model trajectory $H_u$. In decentralized networks characterized by massive deep learning model parameters and numerous communication rounds, requiring edge nodes to cache a large volume of historical checkpoint models could present significant practical difficulties.

---

> ### Author Rebuttal · Authors · 2026-03-31
>
> We sincerely thank you for your time and helpful suggestions. We provide our detailed responses below.
>
> **Reply to Weaknesses:**
>
> **1:** To further evaluate TRACE-DU on a more complex setting, we added experiments on SVHN with 70 clients. The results are shown below.
>
> **Table 1: Comparison on SVHN dataset under Single-DU**
>
> | **Method** | **Retain Acc(%)** | **Time(s)** |
> |---|---:|---:|
> | Retrain | 72.54 | / |
> | CDU | 87.40 | 26.17 |
> | TRACE-DU | 88.81 | 1.04 |
>
> These results are consistent with our main findings: TRACE-DU preserves strong retain utility while remaining substantially more efficient than alternatives.
>
> **2:** We agree that communication cost remains important in decentralized unlearning. However, the $T_r$ rounds in TRACE-DU are standard decentralized training rounds, not a separate coordination overhead introduced by our framework. More importantly, checkpoint selection improves the utility–efficiency tradeoff: compared with perturbing only at the current model, it yields a better starting point for recovery and thus better utility, while compared with retraining from scratch, it requires substantially fewer recovery rounds and hence lower communication cost. We also added experiments with smaller $T_r$ (see Q3), which further illustrate this tradeoff. If further communication reduction is desired, standard compression techniques can be incorporated into the underlying decentralized training/recovery process, and our unlearning framework is fully compatible with such techniques.
>
> **3:** We agree that storage overhead should be quantified. Our method stores only the retained checkpoint trajectory, with space complexity $O(|H_u|d)$, where $|H_u|$ is the number of stored checkpoints and $d$ is the model dimension. It does *not* require storing every client’s full local trajectory. We provide a quantitative comparison in the response to Q2, showing that the additional storage overhead is much smaller than that of PDUDT.
>
> **Reply to Questions:**
>
> **1:** Our analysis assumes that the mixing matrix at each round is symmetric and doubly stochastic. This remains feasible under dynamic topologies and node dropouts by constructing the round-wise mixing matrix using the **Metropolis–Hastings rule**: for an active edge $(i,j)$ at round $t$, set the edge weight to $\frac{1}{1+\max \(d_i^t,d_j^t\)}$, set the diagonal entry so each row sums to 1, and set all other entries to 0. This yields a symmetric doubly stochastic matrix over the active topology at each round and is standard in decentralized learning/unlearning [1-3]. Therefore, as long as this condition is maintained round-wise, our sensitivity analysis continues to apply. On the empirical side, our current experiments already use a fully decentralized setting with randomly generated client connections.
>
> **2:** TRACE-DU requires storing $O(|H_u|d)$ historical parameters, where $|H_u|$ is the number of retained checkpoints and $d$ is the model dimension. By contrast, PDUDT requires each client to store its own local models and the models received from neighbors across rounds before unlearning, leading to per-client overhead $O(t_uN_{max}d)$, where $N_{max}$ is the maximum number of neighbors. To make this comparison more concrete, we measured the extra storage overhead for historical parameters (size in bytes) in Table 2. In practice, this overhead can be further reduced using checkpoint sparsification, low-rank compression, or sliding-window caching that keeps only recent consensus checkpoints.
>
> **Table 2: Comparison of additional storage overhead**
>
> | Method | Additional Storage (MB) |
> |---|---:|
> | PDUDT | 62.55 |
> | TRACE-DU | 1.21 |
>
> **3:** In the extreme regime, any method based on “perturbation + recovery” may enter repeated injection-recovery cycles. This is a fundamental challenge, not one unique to TRACE-SDU. Our current Seq-DU setting already moves in this direction by issuing requests every 40 rounds, under which TRACE-SDU remains substantially more efficient than the baselines while preserving better retain utility. To further test this regime, we added experiments with the interval reduced from 40 to 20 rounds; the results are provided in https://anonymous.4open.science/r/ICML2026_11531A/. These results show that TRACE-SDU remains stable and efficient even under more frequent requests.
>
> [1] He, Lie, Sai Praneeth Karimireddy, and Martin Jaggi. "Byzantine-robust decentralized learning via clippedgossip." arXiv preprint arXiv:2202.01545 (2022).
>
> [2] Qiao, Jing, et al. "PDUDT: Provable Decentralized Unlearning under Dynamic Topologies." Forty-second International Conference on Machine Learning. 2025.
>
> [3] Zehtabi, Shahryar, et al. "Decentralized Sporadic Federated Learning: A Unified Algorithmic Framework with Convergence Guarantees." ICLR (2025).
>
> We appreciate your detailed comments again and hope we have addressed your concerns. Please let us know if you have any additional questions.

---

> > ### Author Rebuttal · Reviewer_P2MJ · 2026-04-03
> >
> > Thank you to the author for the rebuttal. My concerns have been addressed.

---

### Official Review · Reviewer_pvnf · 2026-03-11

**Soundness:** 4
**Presentation:** 4
**Significance:** 3
**Originality:** 3
**Overall Recommendation:** 5
**Confidence:** 2

**Summary:**

This paper considers an interesting problem which is decentralized machine unlearning, i.e., how to remove the impact of one specific client's data. Their analysis is mainly based on SGD stability and model trajectory to derive a sensitivity bound, i.e., by analyzing the impact of the data from the client to be removed on each step, and derive a bound by inducting on steps $t$. This analysis strategy is meaningful and has the potential to be extended to more general settings. This paper is generally well-written and technically solid. The presentations are very clear and the proofs are easy to follow.

**Compliance With Llm Reviewing Policy:**

Affirmed.

**Final Justification:**

I will keep the rating. My main concerns have been addressed.

**Key Questions For Authors:**

1. How to choose hyperparameters in practice? How sensitive is the algorithm's performance to hyperparameters?
2. How good is algorithm's performance when changing $\alpha$ in Dirichlet distribution so that the data are more heterogeneous?
3. When communication is compressed (e.g.  using Top-K compressor in comunication) which is common in practice, is it possible to derive similar bounds?

**Limitations:**

Yes

**Strengths And Weaknesses:**

Strength:
1. The problem studied is of practical importance. Protecting privacy of each client and avoid propagating of harmful information is an important topic in decentralized learning.
2. The analysis is technically sound. The presentation is very clear.
3. The analysis which is based on trajectory can be extended to solving more problems in the area, i.e., analyzing the robustness of $x_t$ when perturbations are applied to earlier steps is useful in theoretical analysis of decentralized learning.

Weakness:
1. It would be better to have some discussions on tightness of the bounds derived.
2. It would be better to test the algorithms at larger scales (larger datasets, more clients)
3. The experiments focus on CV tasks. It would be better to test the algorithms on text tasks.

---

> ### Author Rebuttal · Authors · 2026-03-31
>
> We thank the reviewer for the positive assessment and constructive suggestions. Below we address each point directly.
>
> **Reply to Weaknesses:**
>
> **1:** Compared with prior bounds that are often dominated by deletion-ratio terms such as $\frac{m}{n}$ multiplied by static problem-dependent constants [1,2], our bound is more adaptive and informative. First, the stability factor is inherited from standard SGD stability analysis and thus aligns with existing tightness guarantees at that level. Second, our bound depends on the local deviation from the consensus model, so it shrinks as training progresses. In this sense, it is tighter and captures the training dynamics more precisely than static bounds. Furthermore, the sensitivity bound directly governs the magnitude of the added noise, which in turn determines the utility of the unlearned model. Our experimental results show that our method achieves superior retain accuracy compared to existing DU methods. This empirically substantiates that our theoretical sensitivity bound is significantly tighter than those established in prior DU works. Deriving matching lower bounds is an interesting open problem that we leave for future work.
>
> **2:** To further test scalability, we added experiments on SVHN with 70 clients. The results are shown below.
>
> **Table 1: Comparison on SVHN dataset under Single-DU**
>
> | **Method** | **Retain Acc(%)** | **Time(s)** |
> |---|---:|---:|
> | Retrain | 72.54 | / |
> | CDU | 87.40 | 26.17 |
> | TRACE-DU | 88.81 | 1.04 |
>
> These results remain consistent with our main findings: TRACE-DU preserves strong retain utility while being substantially more efficient than baselines.
>
> **3:** We added a text-domain experiment using BERT-tiny on AG News under Single-DU.
>
> **Table 2: Comparison on AG News dataset under Single-DU**
>
> | **Method** | **Retain Acc(%)** |
> |---|---:|
> | Retrain | 92.02 |
> | PDUDT | 92.09 |
> | CDU | 87.22 |
> | TRACE-DU | 92.62 |
>
> This result suggests that TRACE-DU also transfers well beyond CV tasks.
>
> **Reply to Questions:**
>
> **1:** We describe the main hyperparameter choices in Section 5.1 and Appendix A.2.1, largely following the baselines for fair comparison. TRACE-DU is not parameter-free, but it is less sensitive to hyperparameter tuning than several prior DU methods. The appendix already includes robustness results under varying noise scales, where TRACE-DU maintains stable retain accuracy. We further added experiments varying $T_r$, $N$, $\alpha$, $\epsilon$, which show similar robustness trends. We provide results in https://anonymous.4open.science/r/ICML26_11531A/ and https://anonymous.4open.science/r/ICML2026_11531B/.
>
> **2:** The main paper reports results under non-IID data with Dirichlet $\alpha=0.3$ and 50 clients. Here We additionally evaluated a more heterogeneous setting with **70 clients** and a different Dirichlet parameter under both Single-DU and Seq-DU; the results are provided in https://anonymous.4open.science/r/ICML2026_11531B/.
>
> **3:** We believe so in principle. Communication compression would introduce an additional aggregation error term into the sensitivity recursion. For unbiased compressors, this would typically appear as an extra controllable residual term; for biased compressors, additional control mechanisms such as error feedback or variance-bias reduction may be needed. More broadly, our framework is orthogonal to communication-efficient decentralized training: if the compressor satisfies a suitable bounded-error property, a trajectory-aware bound of similar form should still be derivable. A formal result would require specifying the compressor model and its assumptions, which we leave for future work.
>
> [1] Sekhari, Ayush, et al. "Remember what you want to forget: Algorithms for machine unlearning." Advances in Neural Information Processing Systems 34 (2021): 18075-18086.
>
> [2] Wu, Hengliang, et al. "Certified Unlearning in Decentralized Federated Learning." arXiv preprint arXiv:2601.06436 (2026).
>
> Thank you again for your time and valuable feedback. If you have further confusions, we are happy to clarify them.

---

> > ### Author Rebuttal · Reviewer_pvnf · 2026-04-01
> >
> > My main concerns have been generally resolved. Newly added experiments are extensive and the results are satisfatory.

---

### Official Review · Reviewer_2URH · 2026-03-12

**Soundness:** 3
**Presentation:** 3
**Significance:** 2
**Originality:** 2
**Overall Recommendation:** 4
**Confidence:** 5

**Summary:**

This paper studies decentralized unlearning. By storing checkpoints throughout training, the paper is able to save computation by retraining from a given checkpoint according to a sensitivity measure. Both theoretical and empirical results are provided.

**Compliance With Llm Reviewing Policy:**

Affirmed.

**Final Justification:**

I would like to thank the authors for their thorough and constructive responses. The technical distinctions between the baseline SIFU and the proposed methodology are now clear. I have no further questions and recommend that the authors integrate the clarifications from this discussion into the final manuscript. Additionally, I suggest including a more comprehensive comparison with SIFU to further highlight the paper's contributions.

**Key Questions For Authors:**

1. Can the authors provide a comprehensive comparison with [Fraboni et al. 2024]?
2. Can the authors provide a more fine-grained characterization of $T_r$?
3. Can the authors provide more ablation studies with respect to privacy budget $\epsilon$, topologies and $\sigma$?

My major concerns are centered around the novelty of the work and the impacts of $T_r$ on the unlearning performance.

**Limitations:**

yes

**Strengths And Weaknesses:**

**Strengths**

* Fully decentralized unlearning is generally under-explored. The paper pioneers the principle-guided solutions on it.
* The paper is generally well-written with a good flow.

**Weaknesses**

* Its strong connection with [Fraboni et al. 2024] and, therefore, the concerns of novelty.
    * [Fraboni et al. 2024] focuses on the master-slave architecture and proposes a novel algorithm, SIFU, with an emphasis on all three contributions mentioned in the paper: (i) tracing sensitivity through trajectory, (ii) unlocking the power of multiple local steps, and (iii) beyond convex analysis.
    * The paper has not been compared with [Fraboni et al. 2024] properly. In fact, it is incorrect to discuss the relationship in centralized machine unlearning (line 102).
    * The proposed algorithm is almost identical to SIFU, although with different choices of parameters tailored to fully decentralized learning. The main contribution of the paper is the design of the proxy Eq. (4) in the fully decentralized settings.
*  Dynamic network topologies. Line 126 mentions "we also consider decentralized networks with dynamic topologies instead of fixed ones." In Appendix A.1, this requirement is fleshed out in Assumption A.1. However, the impacts of spectral norm $\rho$ on the unlearning analysis remain unclear.
* The standard deviation $\sigma$ in line 294 is used without introductions. By combining the discussions in the LHS of line 269 and the RHS of lines 221-222, we can see that $\sigma \ge \sigma (U, c)$. In [Fraboni et al. 2024], $\sigma (U,c)$ is chosen to be the same as $\sigma$.
* Intuitively, this framework works for unlearning because, by selecting a given checkpoint according to the proxy, we can guarantee that the further refined unlearned model satisfies the DP-like unlearning notion (Definition 3.2) with respect to the retrain-from-scratch model through post-processing. The subsequent $T_r$ rounds are used to refine the model quality. From this standpoint, the length of $T_r$ rounds determines the final quality of the unlearned model. Unfortunately, the paper does not have any characterizations for that.
* Experiments.
    * Parameter estimation. In non-convex tasks, it is often challenging to obtain the exact smoothness constant $L$. The paper uses a CNN and ResNet-18 for the non-convex tasks. Unfortunately, the parameter estimation process has not been specified.
    * Network structures. Line 343 says that the connections between clients are randomly generated. This raises a natural concern over the impacts of network topologies on the algorithm performance.
    * The privacy budget is set to be $\epsilon=50$ (line 725), which is quite large. More ablation studies are required to understand the performance of the proposed method.

---

> ### Author Rebuttal · Authors · 2026-03-31
>
> We thank the reviewer for the careful reading and detailed comments. Below we address each concern directly.
>
> **Reply to Weaknesses:**
>
> **1:** We agree that SIFU is a highly relevant prior work and that our original discussion did not position it clearly enough. We will make this comparison explicit. That said, we respectfully disagree that our method is “almost identical” to SIFU. While both methods follow the checkpoint-selection-and-refinement paradigm, our setting is fully decentralized, whereas SIFU is designed for a centralized master-server architecture. This distinction is not cosmetic: in our setting there is no server that can directly track client contributions or perform centralized aggregation, so the key quantities used by SIFU are not directly available. Our main technical contribution is precisely to replace these centralized ingredients with decentralized ones, including the proxy in Eq. (4) and the corresponding sensitivity analysis. We provide a detailed comparison in the reply to Q1.
>
> **2:** Our intent was to allow time-varying mixing matrices $Q^t$ as indicated by the superscript $t$, rather than imposing an additional assumption in A.1. In our analysis, $\rho$ characterizes the mixing behavior of the (possibly time-varying) network, but it does not explicitly enter the final sensitivity bound or unlearning guarantee.
>
> **3:** The reviewer is correct that $\sigma\ge\sigma(U,c)$. Hence, line 294 should use $\sigma=\sigma(U,c)$.
>
> **4:** We agree that $T_r$ mainly affects the final utility after unlearning refinement. Our unlearning guarantee is unchanged by these additional $T_r$ rounds due to post-processing (Eq. (21)). Utility after refinement can be understood through the convergence behavior of the underlying decentralized SGD procedure. We also added an ablation with $T_r=20, 60$ (https://anonymous.4open.science/r/ICML2026_11531A/), which confirms the expected utility-efficiency tradeoff while preserving the unlearning guarantee.
>
> **5.1:** In the non-convex experiments, $L$ is used as a practical calibration hyperparameter rather than an exactly computed smoothness constant. This is standard in practice, since exact estimation of $L$ for CNNs/ResNet-18 is itself difficult. For fairness, all methods use the same $L$.
>
> **5.2:** We added results under multiple network structures; please see the response to Q3.
>
> **5.3:** We added an additional ablation with $\epsilon=20$ under a more heterogeneous setting ($\alpha=0.4$, 70 clients), covering both Single-DU and Seq-DU. We provide results in https://anonymous.4open.science/r/ICML2026_11531B/, which confirm that our method consistently outperforms baselines across different parameter settings.
>
> **Reply to Questions:**
>
> **1:** SIFU assumes a centralized server-client architecture, where the server can directly aggregate updates and trace client contributions. Our method targets fully decentralized peer-to-peer learning, where no such server exists. This structural difference fundamentally changes what information is available and therefore requires different technical tools.
>
> Our novelty relative to SIFU is mainly in two aspects:
>
> (1). **A different decentralized contribution proxy:** In SIFU, contribution tracking is enabled by centralized aggregation. In our setting, this quantity is not directly observable. We therefore introduce the proxy in Eq. (4), derived from decentralized learning dynamics. In particular, the factor $\frac{1}{N-1}$ reflects the asymptotically uniform influence that each client exerts on the others through decentralized mixing, and the local model bias serves as a principled proxy for client-specific contribution.
>
> (2). **A different sensitivity anslysis:** Our sensitivity bound is not adapted from server-side aggregation. Instead, we first control the propagation of perturbations via SGD stability at the local level, and then combine this with neighbor aggregation and mixing-matrix properties to obtain the recursive bound in Eq. (14). This step is specific to decentralized peer-to-peer evolution and is the main technical ingredient behind our unlearning guarantee.
>
> **2:** Please see our response to Weakness 4. In short, $T_r$ affects post-unlearning utility rather than the validity of the unlearning guarantee, which remains unchanged by post-processing. We also added an ablation over $T_r$ as noted above.
>
> **3:** We added an ablation for smaller $\epsilon$, see our response to Weakness 5.3. We also added results on regular graphs with different connectivity levels under both Single-DU and Seq-DU settings, see https://anonymous.4open.science/r/ICML2026_11531C/. For $\sigma$ the appendix already reports ablations over 0.025 to 0.2.
>
> Thank you again for your thoughtful review, please let us know if you have any further concerns.

---

> > ### Author Rebuttal · Reviewer_2URH · 2026-04-03
> >
> > I would like to thank the authors for their detailed and constructive responses. The authors have addressed my concerns regarding experiments. I have follow-up questions for the authors on the comparisons with SIFU.
> > * While I agree with the authors that the distributed protocols are different (SIFU: master-slave, here: fully distributed), SIFU also introduces an observation proxy in their equation (13). If we consider clients with equal data volumes, it will also have an $1/(N-1)$ factor, where $N$ is the number of clients. Also, do the clients need to know the global model $\bar{x}^t$ during unlearning?
> > * The equation (14) shares the same intuition as the equation (29) in SIFU, although in the fully decentralized settings.
> > Therefore, I would like the authors to clarify their technical novelties with respect to SIFU, beyond its applications in fully decentralized learning.

---

> > > ### Author Response · Authors · 2026-04-06
> > >
> > > We thank the reviewer for the valuable follow-up. Our clarifications are provided below.
> > >
> > > **Reply to the follow-up questions:**
> > >
> > > **1.1:** In decentralized settings, the $\frac{1}{N-1}$ factor is derived from the fact that the peer-to-peer influence weights converge exponentially fast to the uniform distribution. Although we mentioned it briefly in lines 200–203 and 590, the underlying mathematical justification was not explicitly detailed in the current draft. To be more specific, we have the following mixing bound to quantify how far the actual peer-to-peer influence at round $t$ deviates from the ideal uniform distribution:
> > > $$
> > > ||\Phi(t,p)e_i-\frac{\mathbf{1}_{N}}{N}||^2 \le \rho^{t-p+1},
> > > $$
> > > where $\rho\in(0,1)$, $\Phi(t,p) := Q^{t} Q^{t-1} \cdots Q^{p}$, ($t \ge p$), $Q^{t}$ denotes the mixing matrix at round $t$, $\mathbf{1}_N\in\mathbb{R}^N$ denotes the all-ones vector and $e_i$ denotes the $i$-th canonical basis vector. Thus, we use a uniform weight $\frac{1}{N-1}$ on the retained $N-1$ clients to approximate the peer-to-peer influence after $t$ iterations.
> > >
> > > Here we clarify that $\frac{1}{N-1}$ utilized in our current analysis is the ideal uniform weight. At finite rounds, the actual influence is approximately uniform and includes a decaying finite-time mixing residual term. This point was not made clearly enough, and the current manuscript presents the well-mixed simplified form. We will include a decaying finite-time mixing term controlled by $\rho$ in the revision, as detailed in our response to Q2.
> > >
> > > **1.2:** Our current formulation requires access to $\bar{x}^t$ at the selected checkpoint. In implementation, it can be maintained via an auxiliary consensus protocol over the network, while the underlying training dynamics still follow standard peer-to-peer decentralized mixing.
> > >
> > > **2:** Eq. (14) in TRACE-DU and Eq. (29) in SIFU are both based on the local SGD updates and the related SGD stability from Lemma 3.7 in [1]. Since our framework also employs multiple rounds of local SGD updates (Algorithm 1, line 283) and both methods are grounded in SGD stability, the local stability component in Eq. (14) is formally analogous to Eq. (29) in SIFU.
> > >
> > > The key technical difference lies in the **sensitivity bound itself**. In SIFU, its sensitivity bound depends on the stability factor and a server-side contribution proxy, without any need to account for neighboring mixing.
> > >
> > > By contrast, in the fully decentralized setting, the technical novelty lies in the integration of peer-to-peer mixing properties into sensitivity analysis. In the current version, the peer-to-peer mixing weights were simplified to a uniform distribution for better presentation. We clarify that the peer-to-peer mixing term admits a more refined characterization, which precisely highlights the difference between the decentralized and centralized settings. Here we introduce an additional decaying mixing residual to capture the finite-round discrepancy from uniform mixing. This residual term does not alter the algorithm procedure or the main unlearning guarantee.
> > >
> > > Concretely, for communication round $p$, we define $X^p:=[x_{1}^{p},\ldots,x_{N}^{p}]\in\mathbb{R}^{d\times N}$. Under the time-varying symmetric doubly stochastic topology, the sensitivity bound with mixing residual takes the form
> > >
> > > $$
> > > \mathcal{S}(t,c)\le \sum_{p=0}^{t-1} G(f_{\mathcal I},\gamma)^{(t-p-1)K}
> > > \left[
> > > \frac{1}{N-1}||x_c^p-\hat{x}^p||_2
> > > +
> > > \frac{\Gamma_p}{N-1}\rho^{\frac{p-s+1}{2}}
> > > \right],
> > > $$
> > >
> > > where $s\le p$, $\hat{x}^p:=X^p\Phi(p,s)e_{i}$ is the finite-time mixing estimate of the consensus model, and $\Gamma_p:=||X^p-\bar{x}^p\mathbf{1}^\top||$ measures the consensus disagreement. Here $\Gamma_p\leq\sqrt{N}\max_{j\in[N]}||x_{j}^{p}-\bar{x}^{p}||_2$, which can be bounded by classical consensus error analysis with minor modifications [2,3]. The specific form depends on the convexity of the loss. The proof follows by replacing the ideal proxy $\frac{1}{N-1}||x_c^p-\bar{x}^p||_2$ with the triangle inequality $||x_c^p-\bar{x}^p||_2 \le ||x_c^p-\hat{x}^p||_2+||\hat{x}^p-\bar{x}^p||_2$, and then controlling the second term via the aforementioned mixing bound. Hence the extra correction decays with the number of mixing rounds and becomes negligible once consensus is sufficiently developed, which explains why the uniform-weight approximation is accurate in the well-mixed regime and recovers our current analysis. We will revise the proof with more detailed discussion to state this explicitly.
> > >
> > > [1] Hardt, Moritz, Ben Recht, and Yoram Singer. "Train faster, generalize better: Stability of stochastic gradient descent." ICML, 2016.
> > >
> > > [2] Lian, Xiangru, et al. "Can decentralized algorithms outperform centralized algorithms? a case study for decentralized parallel stochastic gradient descent." Advances in neural information processing systems 30 (2017).
> > >
> > > [3] Koloskova, Anastasia, et al. "A unified theory of decentralized SGD with changing topology and local updates." ICML, 2020.

---

### Official Review · Reviewer_1kkF · 2026-03-13

**Soundness:** 3
**Presentation:** 3
**Significance:** 3
**Originality:** 2
**Overall Recommendation:** 4
**Confidence:** 3

**Summary:**

In this work, authors propose TRACE-DU, a certifiable decentralized unlearing framework that leverages local SGD updates and decentralized model training dynamics. The main contribution is the generic nature of the proposed approach. Namely, TRACE-DU does not require any specific convexity assumptions and does not depend on algorithm hyperparameters. Essentially, authors compute an upper bound on the model sensitivity which is then utilized to determine an unlearning checkpoint as well as to design the model perturbation. Theoretical analysis yields $(\epsilon, \delta)$-unlearning and experimental results demonstrate the effectiveness of the proposed framework. Authors extend the framework to capture sequential unlearning requests as well.

**Compliance With Llm Reviewing Policy:**

Affirmed.

**Final Justification:**

My main concerns have been generally addressed. I will keep my rating.

**Key Questions For Authors:**

1. Why is there a client index $i$ in $G(f_i, \gamma)$ as it seems to be constant across clients according to Table 2? If it is indeed client specific, what is $G(f_{\mathcal{I}}, \gamma)$ in Theorem 4.1?
2. How many independent runs have performed for the experiments?
3. In the implementation details, it is mentioned that sequential unlearning requests arrive at every 40 rounds. Then, in Section 5.2, it is mentioned that you reduce the interval between unlearning requests to a small number of rounds. So what exactly is the number of rounds between subsequent unlearning requests?
4. Does $T_r$ have any effect on the theoretical unlearning guarantee or is it just a hyperparameter chosen to process the unlearning request?

**Limitations:**

No limitation has been discussed. I suggest describing the shortcomings of the work. This could be in the form of future directions or areas for improvement. See the weaknesses above.

**Strengths And Weaknesses:**

Strengths:
1. Unlearning is a well-known and relevant problem. Decentralized nature of the considered scenario presents additional challenges and worth studying. Sequential unlearning capability captures a realistic scenario.
2. The paper is well-written and easy to follow. Often times, authors justify their framework design or implementation choices using the existing literature adding to the credibility of the work.
3. Experiment results support the main theoretical claims of the paper. Detailed elaboration on some of the interesting results is included, particularly in regards to Figure 2.
4. While the paper borrows many familiar techniques from the machine unlearning literature (e.g., model perturbation, divergence between the consensus model and the local model, etc), I find the contribution of the paper still interesting in that TRACE-DU is a generic framework and does not require much additional computation unlike the existing frameworks.

Weaknesses:
1. While TRACE-DU is designed for decentralised unlearning, it does require synchronous operation among the clients.
2. The framework mainly relies on checkpoint computation and perturbation by the client who submits the unlearning request. This would introduce additional security and robustness issues to the system, as there is no central server coordinating the process. For example, a malicious forgetting client can poison the model during this perturbation or send different perturbed models to different clients or dishonestly perturb the model. As the other clients cannot necessarily verify this operation, during the unlearning process, the entire training process for the retained clients may be compromised.
3. While the experiment results cover different areas of the unlearning operation such as the forget accuracy, retain accuracy, MIA performance etc, the paper lacks some key experiment results, including ablation studies. For example, one key component of TRACE-DU is the selection of an earlier checkpoint as opposed to a naive strategy which perturbs the consensus model right after the unlearning request arrives. It would be good to see the performance of this naive strategy to see the exact benefit of this checkpoint operation. Additionally, variables such as $T_r$ and $t_u$ can be varied to demonstrate their effect on the performance of the proposed framework.
4. Some experiments can use additional explanation. For example, in discussing the results in Figure 2, authors mention that retraining may suffer from a generalization issue caused by sequential unlearning requests, thereby disproportionately affecting its performance. While this may be reasonable, even in the case of a single unlearning request, in Fig. 1, we see that retain accuracy of the retraining approach is less than the proposed method. This could use additional elaboration. Another point is that while authors mention that PDUDT requires $T_r$ to be 200, they fix this number as 40 in the experiments. I understand selecting the same $T_r$ in each baseline for fairness, but this number should not favor certain approach over another. In line with my above comment, the paper could use an experiment in which $T_r$ is varied.

---

> ### Author Rebuttal · Authors · 2026-03-31
>
> Thank you for your time and valuable suggestions. Below are our point-by-point responses.
>
> **Reply to Weaknesses:**
>
> **1:** TRACE-DU is built on synchronous decentralized SGD, consistent with standard decentralized optimization settings [1,2,5]. This is a modeling assumption of the underlying training protocol, not an additional requirement introduced by unlearning. In particular, TRACE-DU does not require extra synchronization beyond that already used in decentralized training.
>
> **2:** Our current framework, like prior DU methods (e.g., PDUDT, RR-DU, CDU), assumes honest-but-curious participants that follow the protocol correctly. Under adversarial clients, issues such as dishonest perturbation or inconsistent broadcasts can indeed arise, but these are not specific to TRACE-DU and are common to collaborative/decentralized learning more broadly. In practice, such risks can be mitigated with standard mechanisms such as authenticated broadcast and secure aggregation [3,4,6]. Extending DU guarantees beyond the honest-but-curious setting is an important direction for future work.
>
> **3.1:** We added a comparison under Single-DU on F-MNIST:
>
> **Table 1: Comparison with the naive strategy on F-MNIST dataset under Single-DU**
>
> | **Method** | **Retain Acc(%)** |
> |---|---:|
> | Naive Perturbation | 77.67 |
> | TRACE-DU | 89.71 |
>
> This supports the benefit of selecting an earlier checkpoint rather than perturbing the consensus model directly at $t_u$.
>
> **3.2:** We added ablations with $t_u=40$, $T_r=20$ and $t_u=40$, $T_r=60$ under both Single-DU and Seq-DU. See: https://anonymous.4open.science/r/ICML2026_11531A/.
>
> **4.1 (Retrain vs. TRACE-DU in retain accuracy.):** Even for a single request, Retrain starts from scratch using only retained data, whereas TRACE-DU resumes from an intermediate checkpoint that already preserves useful optimization progress. Under the same recovery budget $T_r$, it is therefore not surprising that TRACE-DU achieves higher retain accuracy.
>
> **4.2 (Choice of $T_r$ in experiments.):** We fixed $T_r$ across methods for a fair comparison under the same recovery budget. We agree that varying $T_r$ is informative, and we added additional studies over $T_r$ as noted above. More broadly, very large $T_r$ values (e.g., 200) mainly allow all methods to recover utility through prolonged retraining, which can obscure differences in the unlearning mechanism itself.
>
> **Reply to Questions:**
>
> **1:** $G(f_i,\gamma)$ denotes the stability factor associated with client-$i$'s local loss and the learning rate. Under our smoothness assumption and shared learning rate, this quantity becomes identical across clients in the analysis, which is why we write $G(f_{\mathcal{I}}, \gamma)$ in Theorem 4.1. We will revise the notation to make this explicit.
>
> **2:** All experiments were averaged over 3 independent runs with different random seeds.
>
> **3:** The interval is 40 rounds. In Section 5.2, “small” refers to being small relative to the 200-round budget used by baselines such as PDUDT. We will state this explicitly.
>
> **4:** $T_r$ serves as a hyperparameter and  only controls post-unlearning utility recovery. The unlearning guarantee is unaffected because the subsequent training is post-processing and therefore preserves the guarantee (line 687).
>
> [1] Lian, Xiangru, et al. "Can decentralized algorithms outperform centralized algorithms? a case study for decentralized parallel stochastic gradient descent." Advances in neural information processing systems 30 (2017).
>
> [2] Koloskova, Anastasia, et al. "A unified theory of decentralized SGD with changing topology and local updates." International conference on machine learning. PMLR, 2020.
>
> [3] Pasquini, Dario, Danilo Francati, and Giuseppe Ateniese. "Eluding secure aggregation in federated learning via model inconsistency." Proceedings of the 2022 ACM SIGSAC Conference on computer and communications security. 2022.
>
> [4] Mansouri, Mohamad, et al. "Sok: Secure aggregation based on cryptographic schemes for federated learning." Proceedings on Privacy Enhancing Technologies (2023).
>
> [5] Giladi, Niv, et al. "DropCompute: simple and more robust distributed synchronous training via compute variance reduction." Advances in Neural Information Processing Systems 36 (2023): 48403-48416.
>
> [6] Nguyen, Truong Son, Tancrède Lepoint, and Ni Trieu. "Mario: multi-round multiple-aggregator secure aggregation with robustness against malicious actors." 2025 IEEE 10th European Symposium on Security and Privacy (EuroS&P). IEEE, 2025.
>
> Thank you again for your valuable comments and hope we have addressed your concerns. If you have further confusions, we are happy to clarify them.

---

> > ### Author Rebuttal · Reviewer_1kkF · 2026-04-02
> >
> > My concerns have generally been addressed. Thank you for the rebuttal.

---

### Decision · Program_Chairs · 2026-04-30

**Decision:**

Accept (regular)

**Comment:**

This paper studies an important problem of certified decentralized unlearning. The key idea is to leverage local SGD updates and decentralized training dynamics to compute fine-grained sensitivity bounds. The reviewers point out several unsolved issues of the current version including synchronous operation assumption, significant communication overhead by high-frequency unlearning, and storage overhead by historical checkpoints. I recommend the authors to take the reviewers' suggestions into account to further improve their work.